# Manual wheelchair biomechanics while overcoming various environmental barriers: A systematic review

Théo Rouvier [1]*, Aude Louessard [1], Emeline Simonetti[1,2], Samuel Hybois[3], Joseph Bascou[1,2], Charles Pontonnier[4], Hélène Pillet[1], Christophe Sauret[1,2]

**1** Institut de Biomécanique Humaine Georges Charpak, Arts et Métiers Institute of Technology, Paris, France, **2** Centre d'Études et de Recherche sur l'Appareillage des Handicapés, Institution Nationale des Invalides, Créteil, France, **3** Complexité Innovation Activités Motrices et Sportives, Faculté des Sciences du Sport, Université Paris-Saclay, Orsay, France, **4** Université de Rennes, Centre National de la Recherche Scientifique, Institut National de Recherche en Informatique et en Automatique, Institut de Recherche en Informatique et Systèmes Aléatoires–Unité Mixte de Recherche 6074, Rennes, France

* theo.rouvier@ensam.eu

**Data Availability Statement:** All relevant data are within the paper and its Supporting Information files.

## Abstract

During manual wheelchair (MWC) locomotion, the user's upper limbs are subject to heavy stresses and fatigue because the upper body is permanently engaged to propel the MWC. These stresses and fatigue vary according to the environmental barriers encountered outdoors along a given path. This study aimed at conducting a systematic review of the literature assessing the biomechanics of MWC users crossing various situations, which represent physical environmental barriers. Through a systematic search on PubMed, 34 articles were selected and classified according to the investigated environmental barriers: slope; cross-slope; curb; and ground type. For each barrier, biomechanical parameters were divided into four categories: spatiotemporal parameters; kinematics; kinetics; and muscle activity. All results from the different studies were gathered, including numerical data, and assessed with respect to the methodology used in each study. This review sheds light on the fact that certain situations (cross-slopes and curbs) or parameters (kinematics) have scarcely been studied, and that a wider set of situations should be studied. Five recommendations were made at the end of this review process to standardize the procedure when reporting materials, methods, and results for the study of biomechanics of any environmental barrier encountered in MWC locomotion: (i) effectively reporting barriers' lengths, grades, or heights; (ii) striving for standardization or a report of the approach conditions of the barrier, such as velocity, especially on curbs; (iii) reporting the configuration of the used MWC, and if it was fitted to the subject's morphology; (iv) reporting rotation sequences for the expression of moments and kinematics, and when used, the definition of the musculoskeletal model; lastly (v) when possible, reporting measurement uncertainties and model reconstruction errors.

**Funding:** This study was funded by the French National Research Agency, "Agence Nationale de la Recherche" (https://anr.fr/) by a grant awarded to the following authors: T.R., A.L., J.B., C.P., H.P., C. S. Grant number: ANR-19-CE19-0007. The funders had no role in study design, data collection and analysis, decision to publish, or preparation of the manuscript.

**Competing interests:** The authors have declared that no competing interests exist.

## 1. Introduction

In 2019, it was estimated that 75 million people in the world require a manual wheelchair (MWC) [1]. MWC users daily face physical environmental barriers such as slopes, cross-slopes, curbs, and uneven terrain that affect their access to buildings and urban areas. Yet, accessibility for people with disabilities is crucial for their social and professional integration [2–4]. Standards and regulations have been established to impose some architectural rules to make public buildings and squares accessible to everyone. However, the regulations are mainly based on the aspects of required space and maximum slope inclination [5]. Despite the improvement of the overall accessibility of public areas, these regulations remain unsatisfactory for a large proportion of MWC users [5–7].

The limitations imposed by environmental barriers in MWC locomotion can be described using the International Classification of Functioning, Disability, and Health (ICF) [8]. The ICF is a framework for describing "dynamic interactions between a person's health condition, environmental factors, and personal factors" [8]. The ICF can therefore be used to identify key elements that need to be addressed in rehabilitation [9, 10], to guide the classification of assistive technology [9], or even to determine the relationship linking wheelchair skills and capabilities with participation frequency and mobility [11]. From that, previous studies have, in particular, revealed the need for better training in overcoming environmental barriers [10]. In addition, the ICF framework could be used by clinicians to adapt MWC training programs according to their patients' capabilities and life projects [12]. To that end, it appears necessary to be able to associate a barrier's difficulty with the user's capabilities. This could be achieved by the quantification and comparison of the physical demands associated with the various environmental barriers encountered.

Biomechanical analysis of locomotion is a reference method to investigate physical demands associated with MWC locomotion. Such biomechanical analysis classically includes the quantification of joint motion and intersegmental loads (forces and torques). Thus, several studies have investigated the physical demands of MWC propulsion when crossing various environmental barriers from a biomechanical point of view [13–17]. Illustrations of environmental barriers that were recreated in a laboratory to that end can be found Fig 1. However, in general, only one type of barrier was investigated in each study, and it appears that no study investigated more than two types of obstacles, hindering the comparison of results between barriers. Moreover, studies seem to use a variety of experimental protocols and investigated different biomechanical parameters. For these reasons, researchers may encounter difficulties when looking for concise data on the influence of environmental barriers on a biomechanical evaluation of MWC locomotion. To address this gap, the purpose of this study was to identify and synthesize data and experimental methods from the literature on the biomechanics of MWC propulsion for various and frequent environmental barriers that are encountered daily by MWC users.

## 2. Methods

The present study conducted a systematic review to identify and analyze existing studies that reported biomechanical parameters of MWC propulsion while overcoming environmental barriers. Because handrim propulsion is the most frequent system of manual propulsion adopted by MWC users due to its higher compliance with the constraints of activities of daily living indoors [18–21], the review focuses on the biomechanics of manual handrim propulsion.

### 2.1 Systematic literature review

To answer the question: "What are the biomechanics involved to overcome specific environmental barriers?", a systematic search was performed based on the methodology of Harris

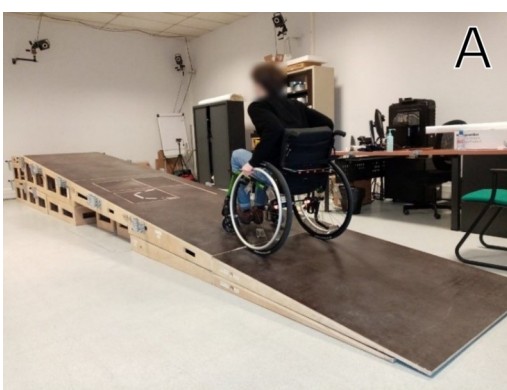

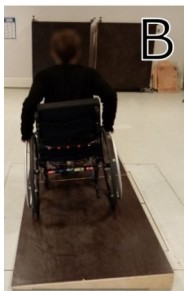 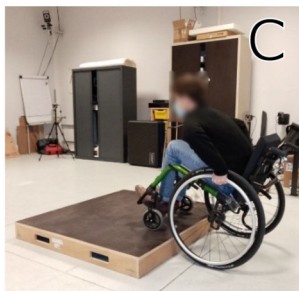

**Fig 1. Reproduction of environmental barriers in a laboratory.** Picture A: reproduction of a slope. Picture B: reproduction of a cross-slope. Picture C: reproduction of a curb.

et al. [22] and Moher et al. [23] to identify relevant articles published until May 2021 within the Pubmed and Scopus database.

The request, launched on May 3, 2021, focused on biomechanical parameters and especially on spatio-temporal parameters, kinematics, kinetics, and muscle activations during MWC propulsion to overcome environmental obstacles, as well as on the experimental methods used to obtain the aforementioned parameters. More precisely, the request was:

*(bioengineering OR biomechanic\* OR kinematic\* OR velocity OR velocities OR (joint angle\*) OR kinetic\* OR force\* OR torque\* OR moment\* OR (motion capture) OR electromyography) AND wheelchair AND (propulsion OR slope OR kerb OR curb OR ground OR floor OR rolling resistance OR activities OR activity OR ambulation OR locomotion OR situation)*

The keywords used for this search were determined after reviewing the results of a preliminary search, which had identified the four most studied in the literature: slope, cross-slope, curb, and ground type.

## 2.2 Article selection

Articles were selected following the flow diagram (Fig 2) recommended by the Preferred Reporting Items for Systematic Reviews and Meta-Analyses (PRISMA) [23]. After eliminating duplicates, all titles were screened for inclusion by three of the authors. The inclusion criteria were: original study or systematic review; study written in English; and features experimental results on slopes, cross-slopes, curbs, and ground types during MWC locomotion. Exclusion criteria were articles about electric wheelchairs, power-assisted wheelchairs, sports wheelchairs, other propulsion systems than manual handrim, and hemiplegia-pattern propulsion. All other abstracts and articles were screened by the same authors. The articles on subject-based studies dealing with an environmental barrier were selected and then sorted according to the barrier type: slope, cross-slope, curb, and ground type.

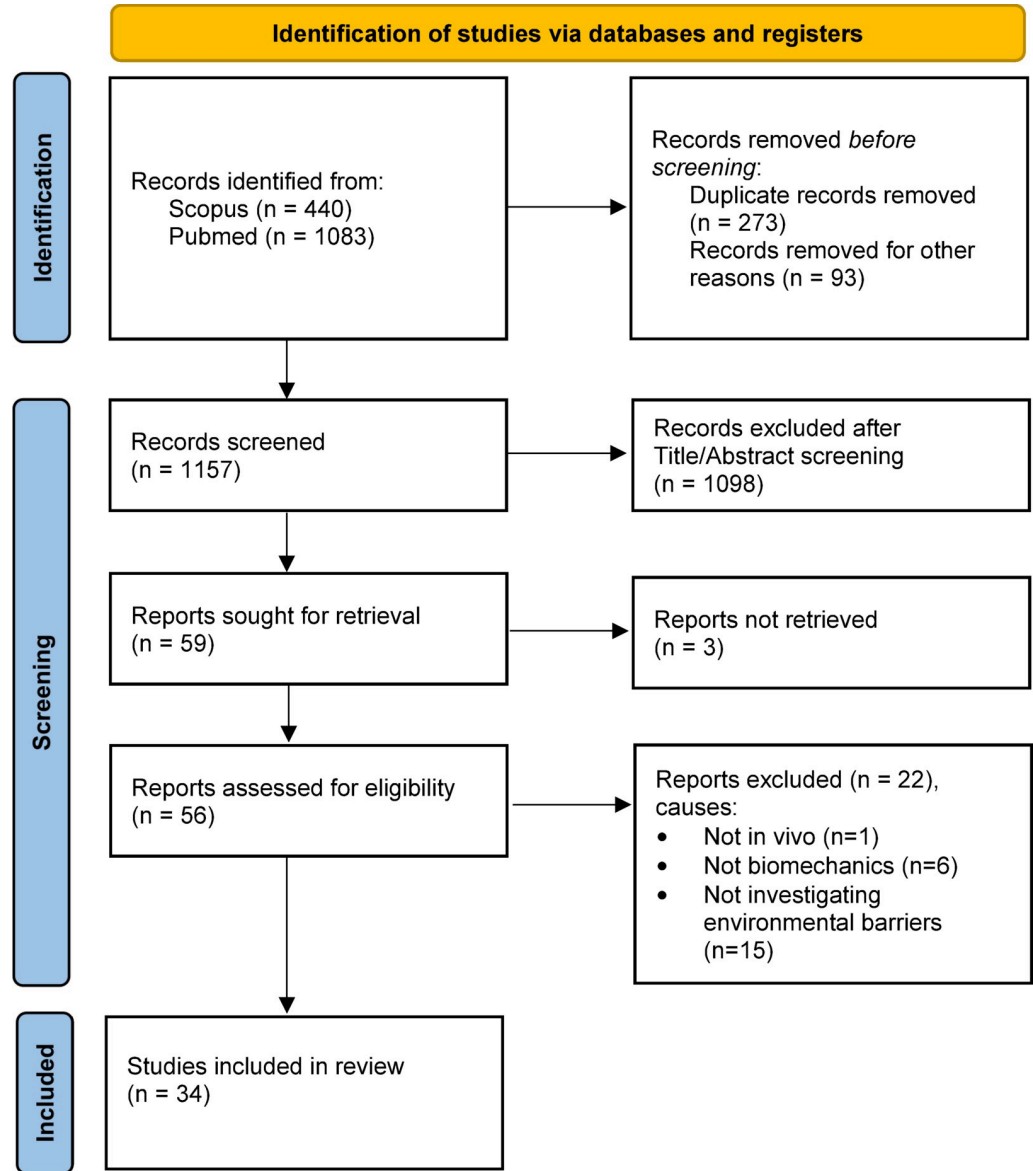

**Fig 2. PRISMA 2020 flow diagram for new systematic reviews which included searches of databases and registers only.**

For the analysis, biomechanical parameters were divided into four *a priori* defined categories: spatio-temporal parameters (push time, recovery time, cycle time, speed, etc); kinematics (joint angles); kinetics (handrim forces and torques, rate of rise, fraction of effective force, net joint moments, mechanical work and power, etc); and muscle activity. A more detailed definition of these biomechanical parameters can be found in S1 Appendix.

## 3. Results

The first search resulted in a total of 1429 references, and 1093 articles remained after removing duplicates. The screening through the title filter resulted in 266 references. After reading the abstracts, 59 articles were selected, and finally, 34 papers were included in this review after the full text read. This selection process is summarized in Fig 2.

The 34 retrieved articles included populations between 7 and 128 participants (Total: 756, Mean [M]: 22, standard-deviation [SD]: 25). Cohorts included able-bodied (AB) subjects and MWC users (MWU), among whom spinal cord injured (SCI) subjects, subjects with lower limb amputation, cerebral palsy, neuropathy, or Friedreich's Ataxia. AB and SCI subjects were studied in 10 and 22 studies, respectively (Table 1).

Experimental design, acquisition methods, and measurement tools were also found to differ between studies. The MWC was propelled overground, on a treadmill, or over a stationary ergometer. Kinematics were recorded either with motion capture systems, inertial measurement units, video cameras or optical encoders. Kinetics were systematically recorded with instrumented wheels.

An overview of the retrieved studies is provided in Table 1. A subsection dedicated to each investigated environmental barrier (slope, cross-slope, curb, ground type) summarizes the experimental methods used in these studies (also reported in Tables 2–5) as well as the obtained biomechanical results. A compilation of the detailed numerical results of the studies is appended as supplementary material (S2 Table).

### 3.1 Slope

**3.1.1 Methods on slopes.**   Twenty-five articles investigated MWC propulsion on a slope, all during slope ascent (*Table 2*). The number of participants ranged between 7 and 128 (M: 23, SD: 29) and the studied populations were mostly MWU (SCI or other motor disabilities).

Experimental design differed across studies, both in terms of the propulsion experimental environment (overground, treadmill, or on a stationary ergometer) and the slope (grade mostly between 2˚ and 5˚ but could reach up to 15˚) (Table 2). Similarly, the acquisition methods and measurement tools were not consistent between studies. Kinematics recording was most often based on opto-electronic motion capture systems, but also on systems based on inertial measurement units, simple 2D cameras, or optical encoders. Kinetics were always measured through instrumented wheels (six-component dynamometers), generally mounted on one side only. One study investigated the kinetics of both wheels using only one instrumented wheel mounted separately on the right and left sides in different trials [49]. Two of the ten studies which used only one instrumented wheel reported having mounted a matching "dummy" wheel on the opposite side to ensure inertial symmetry [7, 13].

Outcome measurements included spatio-temporal parameters (*e.g.* MWC mean velocity, cycle frequency, push and recovery phases durations, etc.), kinematics (glenohumeral, elbow, neck, and trunk angles), handrim kinetics (tangential, radial, and total forces; fraction of effective force, mechanical work and power), joint kinetics (shoulder net joint moments and glenohumeral joint contact force), and muscle activity (percentage of maximal voluntary isometric contraction).

**3.1.2 Results on slopes.**   *3.1.2.1 Spatio-temporal parameters.* Under uncontrolled conditions (*i.e.* overground), MWC speed was found to decrease with increasing slope. Contradictory results were obtained on cycle frequency: MWU tended to increase their cycle frequency with slope on a long ramp [24], whereas AB decreased cycle frequency with slope on a short ramp [7, 48]. Moreover, when the MWC speed was constant across the different slope inclinations (speed imposed by the treadmill belt), cycle frequency tended to increase with increasing slope in SCI subjects [28, 29], but was not affected with AB subjects [13]. Push phase duration at the reference level (*i.e.* grade = 0˚) was similar in all studies that reported this information [24, 28, 29, 36, 45, 48–50]. When the speed was imposed (*i.e.* on a motorized treadmill), the push phase duration was not modified [13, 28, 29] by slope. On the opposite, in overground and stationary ergometer studies, where speed was self-selected, push phase duration increased with the grade [7, 24, 30, 36, 45, 48, 49]. All studies reported a decrease in recovery phase

**Table 1. Synthesis of all studies.**

| Reference | Ground types | Slope | Cross-Slope | Curb | Able-bodied | MWC Users | Video camera | opto-electronic motion capture | IMUs [1] | optical encoder | Instrumented wheel | Spatio-temporal parameters | Kinematics | Handrim Kinetics | Body kinetics | EMG [2] |
|---|---|---|---|---|---|---|---|---|---|---|---|---|---|---|---|---|
| Bertocci et al., 2019 [7] | | x | | | 7 | | x | | | | x | x | | x | | |
| Chow et al., 2009 [24] | | x | | | | 9 | x | | | | | x | | | | x |
| Cowan et al., 2008 [25] | x | x | | | | 128 | | | | | x | x | | x | | |
| Cowan et al., 2009 [26] | x | | | | | 52 | | | | | x | x | | x | | |
| Dysterheft et al., 2015 [27] | x | | | | | 10 | | | | | x | x | | x | | |
| Gagnon et al., 2014 [28] | | x | | | | 18 | | x | | | x | x | | x | | |
| Gagnon et al., 2015 [29] | | x | | | | 18 | | x | | | x | x | x | | x | x |
| Holloway et al., 2015 [30] | | x | x | | | 7 | | | x | | x | | | x | x | x |
| Hurd et al., 2008 [15] | x | | x | | | 12 | | | | | x | x | | x | | |
| Hurd et al., 2008 [31] | x | | | | | 14 | | | | | x | x | | x | | |
| Hurd et al., 2009 [32] | x | x | | | | 13 | | | | | | x | | x | | |
| Kim et al., 2014 [33] | | x | | | 30 | | | | | | | x | | | | x |
| Koontz et al., 2005 [34] | x | | | | | 11 | | | | | x | x | | x | | |
| Koontz et al., 2009 [35] | x | | | | | 28 | | x | | | x | x | | x | | |
| Kulig et al., 1998 [36] | | x | | | | 17 | | x | | | x | x | | | x | |
| Lalumiere et al., 2013 [37] | | | | x | | 15 | | x | | | x | | x | | x | x |
| Levy et al., 2004 [38] | x | x | | | | 11 | | | | | | | | | | x |
| Martin-Lemoyne et al., 2020 [39] | x | | | | | 13 | | | | | | x | | | | x |
| Morrow et al., 2010 [40] | | x | | | | 12 | | x | | | x | | | | x | |
| Morrow et al., 2011 [41] | | x | | | | 12 | | x | | | | | x | | | |
| Mulroy et al., 2005 [42] | | x | | | | 13 | | x | | | x | x | | | x | |
| Newsam et al., 1996 [43] | x | x | | | | 70 | | | | x | x | x | | | | |
| Oliveira et al., 2019 [44] | x | x | | | | 7 | | | x | | x | x | x | x | | |
| Qi et al., 2013 [45] | | x | | | 15 | | | | | | x | x | | x | x | x |
| Requejo et al., 2008 [46] | | x | | | | 20 | | x | | | x | x | | | | x |
| Richter et al 2007 [47] | | | x | | | 25 | | x | | | x | x | | x | | |
| Slavens et al., 2019 [48] | | x | | | 14 | | | x | | | | x | x | | | x |
| Soltau et al., 2015 [49] | | x | | | | 80 | | x | | | x | x | x | x | | |
| Symonds et al., 2016 [50] | | x | x | | 6 | 7 | | | x | | x | x | x | | | x |
| van der Woude et al., 1989 [51] | | x | | | 6 | 6 | x | | | | | x | | x | | |
| van drongelen et al., 2005 [52] | | x | | x | 5 | 12 | | | | | x | x | | x | x | |
| van drongelen et al., 2013 [13] | | x | | | 12 | | | x | | | x | x | | x | | |
| Veeger et al., 1998 [53] | | x | | | 5 | 4 | | x | | | | | x | | | |
| Wieczorek et al., 2020 [54] | | x | | | 8 | | | | | x | | x | | | | x |

[1] IMU: Inertial Measurement Unit;

[2] EMG: Electromyography.

duration with the increase of slope inclination. Seven studies [13, 25, 28, 29, 49–51] reported data on contact angles. Four of these studies used treadmills but investigated different populations, namely AB and MWU, and highlighted significant differences between those populations in contact angle even on a zero grade slope [13, 28, 29, 51]: contact angle was higher on

**Table 2. Study review for slope investigation.**

| Reference | Population | | Experimental condition | Speed | slope grade (°) and length (m) | Kinematics | Kinetics | Muscle activity | Model |
|---|---|---|---|---|---|---|---|---|---|
| **Slavens et al., 2019** [48] | 14 (7F, 7M) | AB[1] | overground | self selected | 0° (10 m) and 4.8° (2.5 m) | opto-electronic (15 cameras, 120 Hz) | | EMG[2] (3 muscles) | Schnorenberg et al., 2014 [55] |
| **Bertocci et al., 2019** [7] | 7 (2F, 5M) | AB | overground | self selected | 3.5, 9.8, 15° (1.22 m) | 1 video camera (30 Hz) | instrumented wheel (dominant side) | | |
| **Holloway et al., 2015** [30] | 7 (7M) | SCI[3] | overground | self selected | 0, 3.7, 6.8° (lengths not reported) | IMU[4] (50 Hz) | instrumented wheel (side not reported) | EMG (3 muscles) | 'Dynamic Arms 2013' (Holzbaur et al., 2005 [56]) |
| **Gagnon et al., 2015** [29] **Gagnon et al., 2014** [28] | 18 (1F, 17M) | SCI | motorized treadmill | self selected (but identical for all slopes) | 0, 2.7, 3.6, 4.8, 7.1° (length: N/A) | opto-electronic (4 cameras, 30 Hz) | instrumented wheels (both sides) | EMG (4 muscles) | ISB Recommendation (Wu et al., 2005 [57]) adapted for shoulder sequence (Senk et Chèze 2006 [58]) |
| **Qi et al., 2013** [45] | 15 (7F, 8M) | AB | overground | self selected | 4° (4.1 m) | | instrumented wheel (side not reported) | EMG (7 muscles) | |
| **van drongelen et al., 2013** [13] | 12 (12M) | AB | motorized treadmill | imposed (1.1 m/s) | 0.6, 1.4, 2.3° (length: N/A) | opto-electronic (6 cameras, 100 Hz) | instrumented wheel (left side) | | only measurement of the hand marker |
| **Chow et al., 2009** [24] | 10 (10M) | 5 SCI, 5 with various disabilities | overground | self selected (normal and fast speed) | 0, 2, 4, 6, 8, 10, 12° (7.3 m) | 1 video camera (60 Hz) | | EMG (6 muscles) | 2D analysis |
| **Oliveira et al., 2019** [44] | 8 (1F, 7M) | 4 SCI, 3 Cerebral palsy, 1 Friedrich's Ataxia | overground | self selected | 0° (10m) and slope with non-constant grade (max grade: 5°, total length: 4.8m) | IMU (11 sensors, 60 Hz) | instrumented wheel (right side) | | Xsens MVN Biomech model |
| **Morrow et al., 2010** [40] | 12 (1F, 11M) | 11 SCI, 1 spina bifida | overground | not reported | 0° and 4.6° (length: 10 m) | opto-electronic (10 cameras, 240 Hz) | instrumented wheels (both sides) | | ISB recommendations (Wu et al., 2005 [57]) |
| **van drongelen et al., 2005** [52] | 17 | 12 SCI, 5 AB | motorized treadmill | imposed (0.56 m/s) | 0° and 1.7° (length: N/A) | opto-electronic (3 cameras, 100Hz) | instrumented wheel (right side) | | Delft Shoulder and Elbow Model |
| **Veeger et al., 1998** [53] | 9 | 4 SCI, 5 AB | motorized treadmill | imposed (0.83, 1.11, 1.39 m/s) | 0.6, 1.1, 1.7° (length: N/A) | opto-electronic (60Hz) | | EMG (1 muscle group) | |
| **van der Woude et al., 1989** [51] | 12 (12M) | 6 MWU[5], 6 AB | motorized treadmill | imposed (0.55, 0.83, 1.11, 1.39 m/s) | 1, 2° (length: N/A) | 1 video camera (54 Hz) | | | |
| **Wieczorek et al., 2020** [54] | 8 | AB | overground | self-selected | 4.6° (4m) | incremental encoder (500 steps) | | EMG (4 muscles) | |
| **Symonds et al., 2016** [50] | 13 (1F, 12M) | 7 SCI, 6 AB | overground | self-selected | 0, 3.7, 6.8° (8.4, 7.2, 1.5m) | IMU (50Hz) | instrumented wheel (left side) | EMG (3 muscles) | |
| **Hurd et al., 2009** [32] | 13 (1F, 12M) | SCI | overground | self-selected | 3° (30m) | | instrumented wheels (both sides) | | |

(*Continued*)

**Table 2.** (Continued)

| Reference | Population | | Experimental condition | Speed | slope grade (°) and length (m) | Kinematics | Kinetics | Muscle activity | Model |
|---|---|---|---|---|---|---|---|---|---|
| **Kim et al., 2014** [33] | 30 (19F, 11M) | AB | overground | self-selected | 4.1, 4.8, 5.7, 7.1, 9.4° (0.9, 1.2, 1.5, 1.8, 2.1, 2.4, 3, 3.6, 4.2, 2.7, 3.6, 4.5, 5.4, 6.3m) | | | | |
| **Kulig et al., 1998** [36] | 17 (17M) | SCI | stationnary ergometer | self-selected | 0, 4,6° (length: N/A) | opto-electronic (50Hz) | instrumented wheel (right side) | | 4 rigid bodies linked by 3 degrees of freedom joints |
| **Levy et al., 2004** [38] | 11 (3F 8M) | MWU | overground | self-selected | 0, 5° (100m, 9m) | | | EMG (8 muscles) | |
| **Morrow et al., 2011** [41] | 12 (1F 11M) | 11 SCI, 1 spina bifida | overground | self-selected | 0° and 4.6° (10 m) | opto-electronic (10 cameras, 240 Hz) | | | ISB recommendations (Wu et al., 2005 [57]) |
| **Requejo et al., 2008** [46] | 20 (20M) | 12 Tetra, 8 Para | stationnary ergometer | self-selected | 0, 2.3, 4.6° (length: N/A) | opto-electronic (6 cameras, 50 Hz) | | EMG (4 muscles) | |
| **Cowan et al., 2008** [25] | 128 (102 M, 26 F) | SCI (various levels) | overground | self-selected | 0, max 5° | | instrumented wheel (side not reported) | | |
| **Mulroy et al., 2005** [42] | 13 (13 M) | SCI | stationnary ergometer | self-selectect | 0, 4.6° (length: N/A) | optoelectronic moetio capture (6 cameras, 50 Hz) | instrumented wheel (right side) | | Inverse dynamics: Kulig et al., 1998 [36] |
| **Newsam et al., 1996** [43] | 70 (70M) | SCI | stationnary ergometer | self-selected | 0, 2.3, 4.56° (length: N/A) | incremental encoder | | | |
| **Soltau et al., 2015** [49] | 80 (74 M, 6 F) | MWU (paraplegic) | stationnary ergometer | self-selected | 0, 4.6° (length: N/A) | opto-electronic motion capture | instrumented wheels (both sides) | | ISB recommendation (Wu et al., 2005 [57]) |

[1]AB: Able-bodied;

[2]EMG: Electromyography;

[3]SCI: Spinal cord-injured;

[4]IMU: Inertial measurement unit;

[5]MWU: Manual wheelchair user.

the same slope when experimenting on AB subjects, and seemed to remain constant with different grades of slope in AB subjects [13], whilst contact angle tended to decrease with increasing slope in MWU [28, 29]

*3.1.2.2 Joint kinematics.* Important differences can be noted between studies in all degrees of freedom (DoF) of the glenohumeral joint. In particular, the evolution of the glenohumeral flexion-extension range of motion (RoM) with the grade differed with either an increase [29], no observed change [44, 48, 50], or even a decrease for AB users in one study [50]. On the contrary, results on trunk inclination are in agreement between studies with an increase of trunk flexion-extension RoM with the grade [29, 44, 50]. An increase of the neck extension with the grade, consistent with the increase of the trunk extension to keep the gaze orientation, was also observed [44]. Wrist flexion-extension and radio-ulnar deviation RoM also tended to increase [53], as well as elbow flexion-extension and pronation-supination RoM [49]. Finally, one study reported maximal scapular angles (down-up, antero-posterior, and internal-external rotations), showing a decrease in maximal downward and anterior rotations, and an increase in internal rotation with increasing slope [41].

Table 3. Study review for cross-slope investigation.

| Study | Population | | Experimental condition | Speed | slope grade (°), length | Kinematics | Kinetics | Muscle activity | Model |
|---|---|---|---|---|---|---|---|---|---|
| **Holloway et al., 2015** [30] | 7 (7M) | SCI[1] | overground | self-selected | 0, 1.4° (length: not reported) | IMU[2] | Instrumented wheel (left side, 50Hz) | Surface EMG[3] (3 Muscles) | Holzbaur et al., 2005 [56] |
| **Richter et al 2007** [14] | 25 (NA) | MWU[4] | motorized treadmill | self-selected | 0, 3, 6° (35m*) | Motion capture system (100 Hz) | Instrumented wheel (downhill side, 200Hz) | | |
| **Hurd et al., 2008** [15] | 12 (11M 1F) | SCI | overground | self-selected | 2° (length: not reported) | | Instrumented wheel (both sides, 240Hz) | | |
| **Symonds et al., 2016** [50] | 13 (1F, 12M) | 7 SCI, 6 AB[5] | overground | self-selected | 0, 1.4° (8.4m, 7.2m) | IMU (50Hz) | 1 Instrumented wheel (left side) | EMG (3 muscles) | |

[1]SCI: Spinal cord-injured;

[2]IMU: Inertial measurement unit;

[3]EMG: Electromyography;

[4]MWU: Manual wheelchair user;

[5]AB: Able-bodied.

*Data estimated by the authors of this review.

*3.1.2.3 Handrim and joint kinetics.* Results on handrim kinetics show noticeable differences between studies when compared to similar or close grades. However, evolution with grade was consistent between studies with an increase of both mean and peak total force, as well as of its tangential and radial components. The handrim mechanical work and power also increased with the grade. Results on the fraction of effective force were however less clear with a mean value that tended to slightly decrease [28], be maintained [13], or increase [45, 49]. Few and disparate outcome data were provided on joint kinetics during slope ascent. An increase of the mean and peak glenohumeral net joint moment and the peak elbow net joint moment with slope was however reported [29, 36, 41, 42, 45]. Two studies reported data on glenohumeral joint contact forces, which require the assessment of muscle forces through a musculoskeletal model, and found a significant increase of the three components of this force with the slope grade [30, 36].

*3.1.2.4 Muscle activity.* Most studies reported peak EMG value [29, 30, 33, 38, 45, 48, 50, 60], but five studies reported mean EMG activity during propulsion [24, 29, 46, 50, 61]. Although most studies reported normalized muscle activity using maximum voluntary contraction testing, one article reported un-normalized EMG activity as voltage measured by the

Table 4. Study review for curb investigation.

| Study | Population | | curb height (cm) | Kinematics | Kinetics | Muscle activity | Model |
|---|---|---|---|---|---|---|---|
| **Lalumiere et al., 2013** [37] | 15 (14M 1F) | SCI[1] | 4, 8, 12cm | opto-electronic (4 cameras, 30Hz) | instrumented wheels (both sides, 240Hz) | Surface EMG[2] (4 muscles) | Desroches et al., 2010 [59] |
| **van Drongelen et al., 2005** [52] | 5 | SCI | 10cm | opto-electronic (3 cameras, 100Hz) | instrumented wheel (left side) | | Delft Shoulder and Elbow Model |

[1]SCI: Spinal cord-injured;

[2]EMG: Electromyography.

**Table 5. Study review for ground type investigation.**

| Study | Population | | Test ground types | Length (m) | Kinematics | Kinetics | Muscle activity |
|---|---|---|---|---|---|---|---|
| **Oliveira et al., 2019** [44] | 8 (7M 1F) | 4 SCI[1], 3 Cerebral palsy, 1 Friedrich's Ataxia | tile; polyfoam mat | 10; 2.2m | IMU[2] (11 sensors, 60Hz) | instrumented wheel (right side, 240Hz) | |
| **Koontz et al., 2009** [35] | 29 (28M 1F) | 25 SCI, 3 lower-limb amputees, 1 Neural palsy | linoleum; carpet | 1.2; 1.5m | opto-electronic (6 cameras, 60Hz) | instrumented wheels (both sides, 240Hz) | |
| **Cowan et al., 2009** [26] | 53 (20M 33F) | MWU[3] | tile; low-pile carpet; high-pile carpet | 12; 7.3; 7.3m | | instrumented wheels (both sides, 240Hz) | |
| **Hurd et al., 2008** [15] | 12 (11M 1F) | SCI | smooth concrete; aggregate concrete; carpet; tile | N/A; N/A; 10; 10m | | instrumented wheels (both sides, 240Hz) | |
| **Hurd et al., 2008** [31] | 14 (12M 2F) | SCI | aggregate concrete; smooth concrete; carpet; tile | 30; 30; 10; 10m | | instrumented wheels (both sides, 240Hz) | |
| **Koontz et al., 2005** [34] | 11 (10M 1F) | 10 SCI, 1 multiple sclerosis, 1 transfemoral amputee | high-pile carpet; low-pile carpet; concrete; pavers; grass; tile; wood | 7.6; 18.3; 15.2; 15.2; 6.1; 15.2; 15.2m | | instrumented wheel (right side, 240Hz) | |
| **Hurd et al., 2009** [32] | 13 (11M 2F) | SCI | smooth concrete; aggregate concrete | 30m | | instrumented wheels (both sides, 240Hz) | |
| **Levy et al., 2004** [38] | 11 (8M 3F) | MWU | linoleum; carpet | 100; 21m | | | EMG[4] (8 muscles) |
| **Cowan et al., 2008** [25] | 128 (102 M, 26 F) hard-tile: 123 low-pile: 94 | SCI (various levels) | hard tile; low-pile carpet | 10; 10m | | instrumented wheel (one side, not reported) | |
| **Dysterheft et al., 2015** [27] | 10 (7 M 3F) | Teenage MWU | tile; carpet; concrete | 15; 15; 15m | | instrumented wheel (both sides, analyzed only at the right side, 240 Hz) | |
| **Martin-Lemoyne et al., 2020** [39] | 13 (9M, 4 F) | SCI | tiled abrasive floor; padded carpet fllor | 10; 10m | | | Surface EMG (4 muscles, dominant arm) |
| **Newsam et al., 1996** [43] | 70 (70M) | SCI | tile; carpet | 15; 12m | optical encoder | Force transducers | |

[1]SCI: Spinal cord-injured;

[2]IMU: Inertial measurement unit;

[3]MWU: Manual wheelchair user;

[4]EMG: Electromyography.

sensor [38]. The muscles investigated in the studies were often different, although most studies measured the muscle activity of the anterior deltoid and pectoralis major [24, 29, 30, 38, 46, 48, 50]. On equivalent slopes, the different studies gave different values of normalized muscle activity for these two muscles. However, it was observed that muscle activity of all of the studied muscles was found to consistently increase with the grade. Some studies reported muscle activity during locomotion higher than the one observed during maximum voluntary contraction testing for some subjects [46, 61].

## 3.2 Cross-slope

**3.2.1 Methods on cross-slopes.** Four articles studied cross-slope propulsion [14, 15, 30, 50] (*Table 3*). Seven to twenty-five (M: 14, SD: 8) MWU—mainly SCI subjects—took part in these experiments. Trials were performed overground [15, 30, 50], or on a treadmill [14],

always at self-selected speeds. Cross-slope inclination ranged between 1.4 and 6˚. Cross-slope length was only reported in one study (7.2 m) [50].

Kinematics recording was based on an opto-electronic motion capture system or on an inertial measurement unit-based system. Kinetics were systematically measured through a six-component instrumented wheel. The downhill side was systematically measured [14, 15, 30, 50], with only one study reporting using a dummy wheel [14], and only one study equipping both wheels [15]. EMG activity of the downhill side was recorded in two studies and focused on three muscles: the pectoralis major, the anterior deltoid, and the infraspinatus [30, 50].

Outcome data were spatio-temporal parameters (MWC speed, cycle frequency, push and recovery phase duration, contact angle), handrim kinetics (tangential and total handrim forces, fraction of effective force, propelling torque, mechanical work, and mechanical power), shoulder joint kinetics (glenohumeral joint contact force) and muscle activity (peak and/or mean of the percentage of maximal voluntary isometric contraction). One study compared the kinetics at the dominant and non-dominant hand sides, while the MWC's right wheel was downside, without investigating the effect of the side of the dominant hand (two participants left-handed) [15].

**3.2.2 Results on cross-slopes.** *3.2.2.1 Spatio-temporal parameters.* The only study reporting data across different grades of cross-slopes showed a decrease of the speed, an increase of the cycle frequency (i.e. decrease of the cycle duration), an increase of the push phase duration, and a decrease of the recovery phase duration with increasing slope [14]. Contact angles on the downhill side did not appear to be affected by the grade of the cross-slope.

*3.2.2.2 Joint kinematics.* The only study investigating body kinematics during cross-slope propulsion found an increase in downhill glenohumeral flexion/extension and internal/external rotation RoM compared to level-ground propulsion [50]. On the contrary, downhill glenohumeral abduction/adduction RoM decreased on the cross-slope and trunk flexion/extension RoM tended to increase only in SCI subjects (and not in AB subjects).

*3.2.2.3 Joint and handrim kinetics.* Peak and mean total forces were shown to increase with increasing grade of the cross-slope [14] or compared to level-ground [14, 30]. The propelling torque on the downhill wheel as well as the mechanical power of this torque were also increased with the grade of the cross-slope. The downhill glenohumeral joint contact force, assessed through a musculoskeletal model, was increased by the cross-slope with respect to level ground in every direction (posterior, superior, medial, and total) [30].

*3.2.2.4 Muscle activity.* Finally, results on downhill side muscle activity based on EMG data showed an increase of mean muscle activity for all investigated muscles during propulsion on a cross-slope compared to level ground for AB and SCI populations [50]; with an increase of peak muscle activity for the anterior deltoid and pectoralis majors, and a decrease of peak activity for the infraspinatus muscle in SCI participants [30].

### 3.3 Curb

**3.3.1 Methods on curbs.** Two studies investigated curb ascent with a MWC [16, 52], involving five and fifteen SCI participants (*Table 4*). Curb height ranged from four to twelve centimeters and curbs were negotiated overground with momentum. Initial instantaneous MWC speed at the beginning of the curb ascent was not reported in any publication.

Kinematics measurements were performed through an opto-electronic motion capture system in both articles but with a small number of cameras for both (less than four). Handrim kinetics were measured using a six-component instrumented wheel, either on one [52] or on both sides [16]. It was not reported whether a dummy wheel was used to equilibrate the MWC when only one instrumented wheel was mounted. EMG data were recorded in one study and

focused on four muscles: biceps, triceps, pectoralis major, and anterior deltoid muscles. Outcome data were trunk inclination and upper-limb joint angles (shoulder, elbow, and wrist joints), upper-limb net joint moments (shoulder, elbow, and wrist joints), and muscle activity.

**3.3.2 Results on curbs.** *3.3.2.1 Joint kinematics.* Reported results on kinematics [16] showed an increase in the RoM of the shoulder and elbow joints with increasing curb height. In general, this increase was related to an increase of the maximal angle value or a decrease of the minimal value of the angle only. The shoulder internal-external rotation RoM was noticeably increased in both the internal and external rotation ranges. Changes in the wrist RoM remained limited in spite of a slight increase of the peak flexion angle. Finally, the trunk inclination was also modified by the curb height with an increase of the RoM and a noticeable increase of the trunk flexion.

*3.3.2.2 Joint and handrim kinetics.* Regarding results on net joint moments, both studies found consistent results for peak total shoulder and elbow moments at high curb level (*i.e.* 10 and 12 cm). Furthermore, peak and mean net shoulder moments were increased for all three moment components, but more especially for the flexion and internal rotation moments. At the elbow, there was also an increase in the total net joint moment, lower than that of the shoulder. The flexion component was the most affected. At the wrist, the increase with curb height was also more limited than at the shoulder and the elbow. The extension and radial deviation components were the most affected. Comparison between joints showed that the higher the initial moment value (*i.e.* at a curb height of four centimeters), the higher the increase. It can also be noticed that extremely high variability (*i.e.* standard deviation) was found in upper-limb joint kinetics.

*3.3.2.3 Muscle activity.* Finally, regarding muscle activity, all four muscles were found to increase their activity with curb height. The biceps brachii and the anterior deltoid muscles appeared to be the most involved between the four studied muscles. Very high variability was also found on these outcome variables.

## 3.4 Ground type

**3.4.1 Methods on ground types.** Twelve studies investigated the influence of various ground types on MWC propulsion [15, 25–27, 31, 32, 34, 35, 38, 39, 43, 44] (*Table 5*). The experiments were conducted on MWU populations ranging from eight to 128 participants (M: 31, SD: 36), among which most were SCI participants. Indoor ground types were mostly studied and one study investigated grass and pavers [34].

Kinematics were recorded using an opto-electronic motion capture system [35] or inertial measurement units [44]. Kinetics were recorded using instrumented wheels mounted on both sides of the MWC [15, 26, 27, 31, 32, 35] or one side only [25, 34, 44]. It was not reported if a dummy wheel was also mounted when only one instrumented wheel was used. Muscle activity was recorded using EMG [38, 39].

Outcome parameters included the spatio-temporal parameters of propulsion (speed, stroke frequency, push phase duration, contact angle), handrim kinetics (tangential, radial, and total handrim forces, fraction of effective force, propelling torque, mechanical work, and power), and EMG data expressed in percentage of maximal voluntary contraction for normalization purposes, or directly as measured in voltage.

**3.4.2 Results on ground types.** *3.4.2.1 Spatio-temporal parameters.* Results showed that self-selected speed was the highest on smooth concrete, tile, and paved grounds, whereas it was the lowest on high-pile carpet, polyfoam mat, grass, and wood grounds [26, 27, 31, 34, 44]. Stroke frequency was the highest on concrete, grass, and paving. High-pile carpets seemed to induce a decrease in speed compared to low-pile carpets [26, 34], and so did aggregate concrete

compared to smooth concrete [31]. In one of two studies, a decrease of stroke frequency was also reported between high-pile and low-pile carpets [26], while in general, similar stroke frequencies were reported for carpet and tile [25, 27, 31, 34, 43].

*3.4.2.2 Joint kinematics*. Regarding the kinematics of upper limbs, results indicated an increase in the RoM of the shoulder, elbow, neck, and trunk during locomotion on a polyfoam mat compared to locomotion on tiles [44].

*3.4.2.3 Joint and handrim kinetics*. The reported results on handrim kinetics showed that propulsion on smooth concrete, tile, and linoleum resulted in the lowest values in peak and mean handrim forces, propelling torque, as well as output work and power [15, 31, 34, 35]. Propulsion on low-pile carpet also presented low values in handrim forces, propelling torque, and output work and power [15, 27, 31, 34]. High-pile carpet, aggregate concrete, polyfoam mat, pavers, and grass were the most constraining ground types with high values in peak, mean, and rate of rise handrim forces, propelling torque, and output work and power, with grass propulsion having the highest of these values [15, 31, 34, 44]. Fraction of effective force was reported in two articles only, and showed propulsion asymmetry between the subjects' dominant and non-dominant sides and presented a high variance among subjects; it was the lowest on smooth concrete, and the highest on grass, as well as generally high on ground types that present higher values in handrim forces and propelling torque [15, 34].

*3.4.2.4 Muscle activity*. Lastly, regarding muscle activity, an increase of the mean activity was found for the anterior deltoid and the triceps brachii from abrasive tile to padded carpet [39], while similar to decreased voltage values were found from linoleum to carpet for these muscles in [38]. Muscle work was also found to double for the anterior deltoid from tile to padded carpet [39].

## 4. Discussion

### 4.1 Investigated environmental barriers

Four different barrier types representing obstacles encountered daily by MWC users were considered and investigated in the literature: slopes; cross-slopes; curbs; and ground types. Among these four barrier types, the slope has been studied the most, always during the ascent, while cross-slopes and curbs (ascent only) were scarcely studied. Yet, the study of curbs and cross-slopes appears particularly relevant since they require specific propulsion strategies. It should be noted that differences in the biomechanics of the uphill and downhill sides during cross-slopes were not investigated.

Out of the thirty-four retrieved studies, nine investigated multiple barriers at once—albeit not more than two [15, 25, 30, 32, 38, 43, 44, 52]. The scarcity of studies on cross-slopes and curbs diminishes the strength of the conclusions drawn by these studies. Indeed, a larger number of studies may have demonstrated contradictory results, as is the case for the retrieved studies on slopes (due to different experimental setups, processing, or populations). The discrepancy of focus between slopes/ground types and curbs/cross-slopes cannot possibly be explained by the lack of cross-slopes or curbs encountered during MWC locomotion in urban areas, since the uneven ground usually encountered may present such environmental barriers, albeit of low grades [2]. Similarly, descending slopes and curbs, or technically challenging situations such as crossing a door threshold with or without a ramp [6] deserve to be studied. For some of these environmental situations, a task analysis could also be considered by separating start-up, propulsion, braking, and turning.

Future studies should therefore be conducted on several different environmental barriers simultaneously, with a special focus on the reproduction of the environments and tasks that are encountered daily by MWUs. Indeed, measuring spatio-temporal parameters, kinematics,

kinetics, and muscle activity using the same methods for all barriers would allow the identification of a set of parameters reflecting the difficulty of any environmental barrier encountered in daily MWC locomotion. Furthermore, to allow for comparison of results between studies, the experimental methods and protocols must be clearly defined and explained. Indeed, the speed of the MWC when approaching a curb strongly influences curb negotiation. Similarly, muscle fatigue may impact how the different barriers are approached, and especially curbs and cross-slopes. Consequently, future research should focus on the standardization of protocols and experimental methods regarding MWC locomotion.

### 4.2 Experimental design

**4.2.1 Studied populations.**   Significant variations were observed in the recruited populations, composed mainly of SCI and AB subjects (twenty-two and ten articles, respectively), but also of lower-limb amputees or subjects affected by cerebral palsy, neuropathy, or Friedreich's Ataxia. Although the level of experience in MWC locomotion has been shown to significantly affect user biomechanics [62], the MWC locomotion skills of the AB subjects were not specified and therefore this may have influenced the results obtained on each environmental barrier. Even when discarding AB subjects, the included MWC users were characterized by various physical conditions, anthropometries, and abilities. While such differences can lead to different propulsion strategies over the same locomotion conditions, it is interesting to have this variety represented in the studied cohorts, to have a representative population of real-world MWC users.

**4.2.2 Reproduction of environmental barriers.**   The difference in the number of studies investigating each barrier may not only be due to a heterogeneous distribution of interest amongst researchers, but also due to practical reasons regarding the methods available to study each barrier. Indeed, researchers can use inclined motorized treadmills or stationary ergometers to simulate slopes and potentially cross-slopes, whereas experiments with curbs and ground types all need to be conducted overground.

Propulsion strategies implemented on a motorized treadmill or a stationary ergometer replicating a slope or cross-slope may differ from those typically used overground. When motorized treadmills were used, the subjects were sometimes secured using safety belts, which were reported to have some looseness in order to limit their influence on the subject's propulsion [14, 28, 29]. Yet, even when secured, the subject may unconsciously fear to fail to sustain the speed of the treadmill and therefore fall, leading to safer propulsion strategies than those that they would have adopted overground. When a stationary ergometer is used, slope simulation is achieved by adding a rolling resistance equivalent to the work needed to ascend the desired slope, sometimes coupled with an incline of the MWC [42, 46]. Yet, the stationary ergometer fails to reproduce the increased risk of wheelchair tipping during slope ascension, as well as the risk of backtracking when an insufficient moment of propulsion is applied to the handrim by the user.

It should also be noted that when using a treadmill, propulsion biomechanics may be impacted by the surface of the treadmill belt which differs from everyday overground surfaces, leading to different strategies over a similar slope. This remark is also valid for different surfaces during overground propulsion on slope and cross-slope.

**4.2.3 MWC configuration.**   MWC configuration is one of the main determining factors when optimizing locomotion for a given user, as it affects propulsion biomechanics as well as other locomotion factors, such as stability [63]. MWC stability, for example, is strongly affected by environmental barriers such as curbs or slopes [62, 64, 65]. Yet, most of the reviewed studied did not report the configuration of the investigated MWC, and those that did

provided only a brief description of the MWC dimensions. The issue lies in the lack of consensus on methodology to characterize and report MWC characteristics/configuration, leading to a major bias limiting the comparison across studies and subjects.

## 4.3 Joint kinematics and kinetics estimation

Upper-limb kinematics and subject kinetics were reported for ascending slope propulsion as well as, to a lesser extent, for curb and cross-slope, but not for ground types. Yet, when reported, methodological differences in kinetic and kinematic acquisition (opto-electronic motion capture system, system based on inertial measurement units) and in data processing (musculoskeletal model used for computation of joint angles and moments [66, 67], point and basis of expression of net joint moments [68, 69]) hinder rigorous comparisons of studies on the same barrier, and prevent the formulation of a reliable evidence-based synthesis of the propulsion biomechanics for each barrier. Lastly, poor data acquisition accuracy may lead to improper conclusions, especially for kinematics and kinetics quantities [70, 71]. This observation may explain some of the contradictory results reported in the studies such as those involving slopes.

When investigating handrim kinetics, all studies used instrumented wheels, but most of them only mounted such wheels on one side of the MWC, whereas mounting them on both sides would enable the comparison of kinetics on each side of the MWC user and the evaluation of possible asymmetries in propulsion strategies. Moreover, only four studies reported the use of a dummy wheel to balance the MWC equipped with one instrumented wheel, which is crucial to ensure natural propulsion strategies. During level-ground propulsion over concrete, which is a situation expected to stress the user symmetrically, a relative difference of 20% between dominant and non-dominant sides of the user was found [15]. The only study that investigated cross-slope locomotion using instrumented wheels on both sides of the subjects' MWC also reported results indicating an asymmetry in handrim forces, propelling torques, mechanical works, and powers when comparing dominant and non-dominant sides of the user [15]. However, they did not report which side was uphill or downhill, which is the most interesting paradigm for interpretation of the results on cross-slopes.

Reported studies also tended to use different musculoskeletal models, yet the definition of joint coordinate systems linked to musculoskeletal models influences both kinematic and kinetics results [72]. Although there is consensus on upper-extremity joint coordinate system definition for kinematics since 2005 [57], the ISB has made recommendation on the reporting of kinetics only as of [73]. Only two studies [30, 36] reported joint contact forces estimations. The reason could be that such a parameter requires a deeper dive into musculoskeletal modeling and simulation because it requires, as a prerequisite, to assess muscle forces [74]. Furthermore, the definition of such a model influences the accuracy with which joint contact forces are estimated [75]. Further studies should take better advantage of musculoskeletal models specifically developed and tailored to study MWC locomotion, and the sharing of these models would favor the standardization of the results.

It should be noted that none of the studies presented in this review reported the uncertainties in the determination of the parameters of interest, while the different choices of models or measurement devices might have resulted in significant uncertainties. For instance, multibody kinematics optimization was found to generally carry reconstruction residual errors on markers ranging from four to forty millimeters, and between three and ten degrees of error against true bone kinematics for shoulder rotations [76]. Moreover, the measurement uncertainty of kinetic measurement devices given by manufacturers has to be applied and propagated with those kinematic uncertainties to rigorously compare results on body kinetics. Therefore, future

studies should provide recommendations on how to assess and propagate modeling and measurement uncertainties in order to allow a more rigorous comparison of results across different studies.

## 4.4 Muscle activity estimation

Fourteen studies reported results acquired using EMG, ten of which focused on slope propulsion. All studies but one normalized EMG data acquired during locomotion by EMG data of maximum voluntary contraction, hence reported muscle activity highly depends on the physical capacity of each participant. It is therefore difficult to give an estimate of activity for a specific muscle and barrier, as these results are highly dependent on both the subject's physiology and propulsion strategy. Moreover, maximum voluntary contraction normalization is subjected to uncertainty under the risk of incorrectly testing for maximum voluntary contraction. In particular, when normalization is done improperly, there may be trials where recorded muscle activity is higher than its maximal value, characterized by results above 100% of maximum voluntary contraction. For instance, Requejo et al. reported mean muscle activity higher than 100% for eight subjects [46], but it might also be the case for some subjects in other studies in which the mean muscle activity was averaged over all the participants. One study reported un-normalized EMG data, which is therefore presented in Volts [38], preventing the comparison of muscle activity with other studies.

## 5. Conclusion

This review highlighted discrepancies in focus given to each environmental situation in the literature. Slope ascent and ground types were studied much more than cross-slope or curb ascent. Furthermore, the review evidences a lack of consensus on the parameters of interest to report and on the methods used to conduct experiments. These variations and lack of consensus make it impossible to cross-reference studies to compare situations. Nevertheless, for each environmental barrier, this review provides an unprecedented overview of its current biomechanical assessment through the report of numerical values of all biomechanical parameters retrieved from the relevant literature (in tables provided in supplementary material).

At the end of this review process, we recommend a more systematic approach when reporting materials, methods, and results for the reflection of the difficulty of any environmental barrier encountered in MWC locomotion: (i) effectively reporting barriers' lengths, grades, or heights; (ii) striving for standardization or a report of the approach conditions of the barrier, such as velocity, especially on curbs; (iii) reporting the configuration of the used MWC, and if it was fitted to the subject's morphology; (iv) reporting rotation sequences for the expression of moments and kinematics, and when used, the definition of the musculoskeletal model; (v) when possible, reporting measurement uncertainties and model reconstruction errors.

## Supporting information

**S1 Appendix. Biomechanical parameters definition.**
(DOCX)

**S1 Table. PRISMA checklist.**
(DOCX)

**S2 Table. Study and results review for slopes, cross-slopes, curbs, and ground types.**
(XLSX)

## Acknowledgments

The authors would like to thank Pr. Philip Fink for his generous help checking this manuscript's grammar and spelling.

## Author Contributions

**Conceptualization:** Christophe Sauret.

**Investigation:** Théo Rouvier, Aude Louessard, Emeline Simonetti.

**Methodology:** Théo Rouvier, Aude Louessard, Christophe Sauret.

**Supervision:** Christophe Sauret.

**Writing – original draft:** Théo Rouvier, Aude Louessard, Emeline Simonetti, Christophe Sauret.

**Writing – review & editing:** Emeline Simonetti, Samuel Hybois, Joseph Bascou, Charles Pontonnier, Hélène Pillet, Christophe Sauret.

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
