## [Decision Letter · Decision Letter 0]

26 Dec 2021

PONE-D-21-13523Manual wheelchair biomechanics while overcoming various environmental barriers: a systematic reviewPLOS ONE

Dear Dr. Rouvier,

Thank you for submitting your manuscript to PLOS ONE. After careful consideration, we feel that it has merit but does not fully meet PLOS ONE’s publication criteria as it currently stands. Therefore, we invite you to submit a revised version of the manuscript that addresses the points raised during the review process.

Apologies for the delayed return of your manuscript and the accompanying reviews, which is fully on my account. The reviewers appreciate the manuscript and the work it has entailed. They are positive, yet have a number of major suggestions to improve the manuscript and its readability. Other than that they have provided extensive smaller remarks. Major points are the overlap between the results text section and the table info. A more complementary text to the central role of the tables is suggested. Use SI-system units throughout the manuscript. What is the role for Power output (W), external or user-related power production in your analyses? I miss this info in the tables. Add definitions The conclusion should be short and concise, not repeating or adding to the discussion. Did the authors conduct a quality assessment of the different papers, which is often typical for the systematic review? Be consistent in terminology, abbreviations between the different tables and sections. Please submit your revised manuscript by Feb 09 2022 11:59PM. If you will need more time than this to complete your revisions, please reply to this message or contact the journal office at plosone@plos.org. Please include the following items when submitting your revised manuscript:A rebuttal letter that responds to each point raised by the academic editor and reviewer(s). You should upload this letter as a separate file labeled 'Response to Reviewers'.A marked-up copy of your manuscript that highlights changes made to the original version. You should upload this as a separate file labeled 'Revised Manuscript with Track Changes'.An unmarked version of your revised paper without tracked changes. You should upload this as a separate file labeled 'Manuscript'.

We look forward to receiving your revised manuscript.

Kind regards,

Lucas van der Woude

Academic Editor

PLOS ONE

Additional Editor Comments (if provided):

Apologies for the delayed return of your manuscript and the accompanying reviews. The reviewers appreciate the manuscript and the work it has entailed. They are positive, yet have a number of major suggestions to improve the manuscript and its readability. Other than that they have provided extensive smaller remarks.

Major points are the overlap between the results text section and the table info. A more complementary text to the central role of the tables is suggested. Use SI-system units throughout the manuscript. What is the role for Power output (W), external or user-related power production in your analyses? I miss this info in the tables. Add definitions The conclusion should be short and concise, not repeating or adding to the discussion. Did the authors conduct a quality assessment of the different papers, which is often typical for the systematic review? Be consistent in terminology, abbreviations between the different tables and sections.

Journal Requirements:

Reviewers' comments:

Reviewer's Responses to Questions

**Comments to the Author**

1. Is the manuscript technically sound, and do the data support the conclusions?

Reviewer #1: Yes

Reviewer #2: Yes

2. Has the statistical analysis been performed appropriately and rigorously? 

Reviewer #1: N/A

Reviewer #2: Yes

3. Have the authors made all data underlying the findings in their manuscript fully available?

Reviewer #1: Yes

Reviewer #2: Yes

4. Is the manuscript presented in an intelligible fashion and written in standard English?

Reviewer #1: Yes

Reviewer #2: Yes

5. Review Comments to the Author

Reviewer #1: This review gives a very good and extensive overview on the existing literature of manual wheelchair propulsion while overcoming different barriers in daily life of wheelchair users. The results are presented in detail in the tables which is very convenient for the interested reader to extract the information of interest. However, the presentation of the results in the text is sometimes lengthy and a repetition of what is presented in the table. This makes the review difficult to read. In my view, the review would benefit from shortening the text in the results section and referring to the tables instead. Detailed comments to the manuscript are written below.

Introduction:

Line 55: delete (MWU)

Line 55-57: This sentence is not clear. What is the link between ICF and physical demands associated with barriers? Please rephrase.

Line 67-68: Sentence is not complete.

Line 69: replace "in the same study" with "in each study".

Line 78-80: these two sentences are not well placed, they should be placed earlier in the introduction not just at the end where it's out of context.

Line 80: to be specific, add "manual" to handrim propulsion

Methods:

Line 83: replace "consisted of" with "consisted in". There are other small language flaws in the manuscript, please let it check by a native English speaker.

Line 95-98: did you also try to include "inclination" as an addition to "slope"? This might have resulted in more studies.

Line 118: S1 Appendix: The appendix does not includ all parameters mentioned here, and not all parameters from the appendix are mentioned here. Please make it consistent.

Regarding the S1 Appendix:

- Contact angle: you define it as "angle distance travelled by the non-dominant hand on the handrim during the push phase". Why is it limited to the non-dominant hand?

- Rate of rise: your definition is not correct, not detailed. Is it the mean resultant force divided by the contact time of the whole cycle?

Besides that, in literature there are different definitions:

• taking the derivative of FR with respect to time and then determining the maximum value during the first third of the stroke

• (first) peak of the resultant force, divided by the time to reach the peak.

Therefore, write it more precisely in your definition.

- Fraction of effective force: There are also different definitions used, either FEF = (Mwheel·r−1)·Ftot−1 or Ftan2/Fres2

Therefore, be more specific in your definition

Results

Line 133: include a reference to table 1.

Table 2: van Drongelen 2005: information on kinematics is missing. I did not check the whole table on missing information, but just spotted this one. So please check again whether everything is included here.

Table 3: - what does * mean?

- what does sEMG mean?

- Since it is a new table, explain the abbreviations below. Same accounts for the following tables.

Table 5: why is in this table height and weight of the participants included, but not in the others? Keep it consistant over the tables.

Line 156 and further: As already indicated in the general comments, all the information you give in here is already displayed in the table. The first part of the methods, where you summarize the range of participant number, the studied population and the range of slopes measured, is ok (although also not really needed since it is already indicated in the table). The second part on how the measurements were performed is obsolete, since you list again what is mentioned in the table (i.e. 19 articles measured kinematics, 15 article measured kinetics,..). Please shorten this part a lot or even delete it and refer to the table. The same accounts for cross-slope, curb and ground types.

Line 157: it should be referenced to table 2 instead of table 1.

Line 187 and further: In general, I like the overview on the results in the appendix, it gives a good overview. If possible it would be good to take the tables out of the appendix and place it in the manuscript Then you could also shorten the text of the results part.

Line 187: Please refer to appendix Table S2, also further on in the results section.

Line 188: replace "speed decreases with the grade of the slope" with "speed decreases with increasing slope"

Line 193: replace "increase with the grade" with "increase with increasing slope"

Line 215: replace "increase in internal rotation with the slope" with "increase in internal rotation during propulsion on a slope"

Line 241: should be referred to table 3 instead of 2.

Line 250: before you always called it instrumented wheel instead of handrim dynanometer. Please use one of the labels conistently throughout the manuscript

Line 270: "on speed, cycle frequency and duration of push and recovery phase" can be deleted. In general, write more concisely and omit the information that is not really necessary or is a repetition. This makes it easier and more convenient for the reader.

Line 274-275: give reference to the study and Appendix

Line 296: should be referred to table 4 instead of 3.

Line 296: "real" MWC sounds odd, please delete.

Line 313: add reference to appendix.

Line 344: should be referred to table 5 instead of 4.

Line 365: this is another example of sentences that are not needed, a reference to the appendix is more meaningful.

Discussion

Line 408: "investigated in literature" instead of "from literature"

Line 409-414: This information belongs to the results, there you already listed how many studies investigated which barrier. It does not have to be mentioned again, especially not with numbers of studies. You might want to say in one sentence what has been studied most.

Line 431: safety belts were not used in all of the studies conducted on treadmills

Line 443: Number of studies reporting kinematics and kinetics is for the results section, not for the discussion.

Line 467: include reference to the study.

References

Reference 1: what kind of a reference is this? if it is a website, please indicate the url.

Reference 11: Title is written twice in the reference

Figure 1: indicate in the figure why the records (n=1098) they are excluded

S2 Table: Check the table, units are not always indicated. For example in Slope: MWC speed, recovery phase duration, contact angle, sometime it's indicated after the numbers instead of in the the title line (i.e. body kinetics), or both in the title line and after numbers (Peak un-normalized EMG). Please make it consistent.

Reviewer #2: Comments to editor:

The authors presented a nicely written systematic review on environmental barriers and manual wheelchair biomechanics. The introduction, rationale and methods of this study are presented in a clear way. The results still have quite some redundant information, which is already present in the tables, it is recommended to shorten parts of the results to improve the readability. The discussion section raises very interesting points, only the structure needs some finetuning. The conclusion should be shorter and to the point. A recommendation of Acceptance with Major Revision has been given.

General comments:

A nicely written systematic review, which presents the work in a very detailed and extensive way. In more detail:

Introduction: the introduction is written well, only the ending could use some finetuning. Methods: Methods section is well presented and only lacks a clear inclusion/exclusion overview.

Results: The results were separated for the four areas (slopes, cross-slopes, curbs and ground type), as well as methods and results.

Regarding the methods on slopes, cross-slopes, curbs and ground type: It is unnecessary to present all results on methods this extensively. Most information can also be found in the presented tables, which already cover a lot of information. Removing parts in these sections might help with the readability of the whole paper.

On the Results on slopes, cross-slopes, curbs and ground type part: Adding subheadings for the subcategories: spatio-temporal, kinematics, kinetics and muscle activity are recommended.

Discussion: In the discussion section a lot of important information is discussed. It feels like a lot of separate interesting discussion points are raised but the structure is a bit lacking. The transitions from one paragraph to another, as well as the general structure should use some finetuning.

Conclusion: In this section a lot of information is discussed, which can either be removed or combined with some of the discussion points. Try to get the conclusion short and to the point.

Smaller general points: The consistency in the use of references, sometimes all references are listed extensively, but there are cases in which some the references are missing, examples are Lines 380-382 and Lines 426-442.

Maybe a figure with the 4 environmental barriers might be a good addition to have a clear overview and explanation of the 4 barriers (cross-slope might be unclear to some naïve readers).

Specific comments:

Abstract:

Page 2, Lines 33-35: Make the conclusion at the end a bit stronger, something similar to: “It is recommended to standardize the procedure, studying a wide set of…”

Introduction:

Page 3, Lines 45-46: The sentence seems confusing, maybe something similar to: “Despite the improvement of overall accessibility of public areas…”

Page 3, Lines 46: Recommended to start a new paragraph when ICF is introduced

Page 3, Lines 61-62: Last sentence of this paragraph feels a bit redundant

Page 4, Line 75: Remove the word “therefore” in this sentence

Page 4, Lines 78-80: This paragraph is unnecessary, the end of the previous paragraph is a more proper ending of the introduction

Methods:

Page 4, Line 83: Change the sentence a bit: “The present study conducted a systematic review to identify…”

Page 5, Lines 105-106: Remove “Appended to this paper…” and add “(S1 table)” add the end of “…(PRISMA) Checklist [24]”

Page 5, Lines 106-110: Separate into inclusion/exclusion criteria, inclusion: biomechanical data, experimental work, English languages, exclusion: electric and power assisted wheelchairs etc.

Page 5, Line 111: Remove the word “three” from the sentence

Page 5, Lines 114-116: Very nice and detailed description of the parameters

Page 6, Line 117: Change towards: “A more detailed definition of the biomechanical parameters can be found in S1 Appendix”

Page 6, Line 118-119: Sentence feels a bit redundant

Results:

Page 6, Lines 129-132: Split the sentence into two

Page 6, Lines 134-137: Cut the sentences a bit, similar to: “Experimental design, acquisition methods and measurement tools differed between studies. The MWC was propelled overground… …ergometer. Kinematics were either recorded…”.

Page 6, Lines 137-138: What is the use of “on the contrary”?

Page 6, Lines 140-142: Try to rephrase the sentence a bit, it is unclear what is part of the tables and what of the supplementary material. Maybe similar to: “Table 1 represents the synthesis of all studies, the remaining tables (2-5) summarize the study review for the four categories (slope, cross-slope, curb, ground type). … are appended as supporting information (S2 Table)”.

Page 18-19, Methods on slopes: See general comment

Page 19, Line 188-189: Sentence can be removed (“Results on cycle frequency…) and combined with the next sentence.

Page 19, Lines 191-194: Try to shorten the sentence: “…increase with the grade in SCI subjects, but unaffected with AB subjects.”

Page 19: Lines 201-203: What kind of differences were found?

Page 20: Line 212: Remove the part “in the study where they were reported”.

Page 20: Lines 219-220: Remove the part: “…in all studies reporting these data”.

Page 21: Line 233: Change to past sense

Page 21-22, Methods on cross-slopes: See general comments

Page 22: Lines 266-267: Is an interpretation sentence, better be part of the discussion

Page 22, Lines 270-274: Try to shorten this sentence, parts at the beginning of the sentence can be removed.

Page 23-24, Methods on curbs: See general comments

Page 23, Line 296: Remove the word “real”

Page 25-26, Methods on ground types: See general comments

Page 27, Lines 380-382: The references are missing

Discussion

Page 27-28, Lines 401-406: This paragraph only repeats the aims, without actually giving any relevant information, combining some of this information with the next paragraph might be a solution.

Page 28, Lines 417-425: This part can be shortened a bit, there is some repetition, especially with the last sentence.

Page 29, Lines 428-430: Future and past sense are both used in this sentence

Page 29-30, Lines 443-454: The paragraph starts with summarizing the number of articles investigating kinematics/kinetics. Despite main focus of this paragraph seems on the differences in acquisition methods. Try to get the most relevant information at the start/end of the paragraph.

Page 30, Lines 469-476: This feels like a separate paragraph, since the main focus of this paragraph was on the use of instrumented measurement wheels on both side of the MWC.

Page 31, Lines 486-490: It is unclear why this sentence is presented in between brackets, would make the argument stronger to present it as an example.

Page 32, Lines 505-508: Try to combine these sentences into one strong sentence.

Page 32, Lines 509-516: This paragraph does not add too much to the whole review and could, feels like lose paragraph at the end of the discussion, see also general comments

Conclusion:

Page 32-34: A conclusion is preferably short and to the point. The conclusion now consists of around five paragraphs in which also a lot of recommendations are given. Some of these recommendations are already discussed previously and don’t necessarily strengthen the paper.

Page 32, Lines 519-524: The first paragraph summarizes/repeats what has already been repeated multiple times

Page 33, Line 526: This is one of the main lines to be used for the conclusion.

Page 33, Lines 530-536: This all can be said in one sentence in the discussion part.

Page 33, Lines 537-540: Interesting point, also combine with some of the paragraphs in the discussion part (Future recommendation)

Page 33, Lines 541-543: Repeating the fact that cross-slopes and curbs should be studies, but now focused on asymmetrical propulsion.

Page 33-34, Lines 544-551: This seems like one of the main parts of the conclusion of the paper

6. PLOS authors have the option to publish the peer review history of their article (what does this mean?). If published, this will include your full peer review and any attached files.

Reviewer #1: **Yes: **Ursina Arnet

Reviewer #2: **Yes: **Riemer Vegter & Thomas Rietveld

---

## [Author Response · Author response to Decision Letter 0]

10 Feb 2022

Dear reviewers, Editor,

Thank you for your encouraging comments while reviewing the manuscript and for allowing us to improve its content. 

Please find below how we have addressed each of the reviewers’ comment. The reviewers’ comments are reported in black, whereas the answers are reported in blue.

A version of the revised manuscript has also been uploaded. Changes are highlighted in yellow in the revised version of the manuscript.

Kindest regards,

Théo Rouvier, on behalf of all the authors of the manuscript entitled "Manual wheelchair biomechanics while overcoming various environmental barriers: a systematic review”

Reviewer 1:

General comments:

This review gives a very good and extensive overview on the existing literature of manual wheelchair propulsion while overcoming different barriers in daily life of wheelchair users. The results are presented in detail in the tables which is very convenient for the interested reader to extract the information of interest. However, the presentation of the results in the text is sometimes lengthy and a repetition of what is presented in the table. This makes the review difficult to read. In my view, the review would benefit from shortening the text in the results section and referring to the tables instead. Detailed comments to the manuscript are written below.

Thank you for your appreciation and for your detailed review of our work. You will find hereafter responses to your detailed comments. Following your advice, the text in the “results” section was shortened and references to the tables were added.

Introduction:

Line 55: delete (MWU) 

“(MWU)” was deleted from line 47 but the abbreviation was redefined at line 130 of the revised manuscript since it is used in the “results” and “discussion” sections.

Line 55-57: This sentence is not clear. What is the link between ICF and physical demands associated with barriers? Please rephrase.

We believe that ICF could be better implemented for MWC users if clinicians were able to adapt training programs in accordance with the difficulty of a barrier with regards to a user’s capabilities. The quantification of the physical demands associated with barriers could be used as a quantification of a barrier’s difficulty. The paragraph was rephrased. (l. 55-60)

Line 67-68: Sentence is not complete.

The sentence was rephrased. 

Line 69: replace "in the same study" with "in each study".

This was corrected (line 67 of the revised manuscript).

Line 78-80: these two sentences are not well placed, they should be placed earlier in the introduction not just at the end where it's out of context.

The sentences have been moved up in the text at the beginning of the “Methods” section (lines 82-86 of the revised manuscript).

Line 80: to be specific, add "manual" to handrim propulsion

The adjective “manual” was added to be more specific, following the reviewer’s comment. (line 86 of the revised manuscript).

Methods:

Line 83: replace "consisted of" with "consisted in". There are other small language flaws in the manuscript, please let it check by a native English speaker.

The sentence was rephrased. The manuscript was checked by an English speaker and language flaws have been corrected throughout the manuscript.

Line 95-98: did you also try to include "inclination" as an addition to "slope"? This might have resulted in more studies.

Following the reviewer’s suggestion, we performed a new search on PubMed with the addition of the term “inclination”, which resulted in 21 new results compared to the initial PubMed search, among which some had already been retrieved with the initial Scopus search. Based on the screening on the title, most are related to seat or backrest inclinations, and not to the ground inclination (slope grade). Finally, only 1 reference appeared to be relevant, but it did not meet the inclusion criteria based on the screening on the abstract. 

We agree with the reviewer that it remains possible that we missed some papers relative to very specific situations due to the absence of very specific keywords, but we believe that the global conclusions drawn on methodologies will not be challenged.

Line 118: S1 Appendix: The appendix does not include all parameters mentioned here, and not all parameters from the appendix are mentioned here. Please make it consistent.

Thank you for your comment, the parameters defined in the appendix and those mentioned in the “methods” section are now consistently the same.

Results:

Line 133: include a reference to table 1.

A reference to table 1 was added.

Table 2: van Drongelen 2005: information on kinematics is missing. I did not check the whole table on missing information, but just spotted this one. So please check again whether everything is included here.

Thank you for noticing this mistake. Information on kinematics recording in van Drongelen 2005 was added to Table 2. All the tables were double checked following your comment.

Table 3: - what does * mean?

 * Represents data that was extrapolated by the authors using the data given by the study. The signification of “*” was added to the tables, in the caption.

- what does sEMG mean?

sEMG means Surface EMG, this was detailed in the tables’ captions.

- Since it is a new table, explain the abbreviations below. Same accounts for the following tables.

All abbreviations are now defined in the caption of each table.

Table 5: why is in this table height and weight of the participants included, but not in the others? Keep it consistant over the tables.

It is true that data reported in tables was not consistent. We initially chose to report height and weight of subjects for the studies focusing on ground types because some of these studies normalized kinetic data on subject height and weight. But it is indeed confusing to report such data only for those studies and we don’t consider it crucial to the comprehension of our systematic review. For these reasons we normalized the tables and removed data on subject height and weight.

Line 156 and further: As already indicated in the general comments, all the information you give in here is already displayed in the table. The first part of the methods, where you summarize the range of participant number, the studied population and the range of slopes measured, is ok (although also not really needed since it is already indicated in the table). The second part on how the measurements were performed is obsolete, since you list again what is mentioned in the table (i.e. 19 articles measured kinematics, 15 article measured kinetics,..). Please shorten this part a lot or even delete it and refer to the table. The same accounts for cross-slope, curb and ground types.

Thank you for your comment. Following advice from both reviewers, the presentation of the experimental methods of the reviewed studies was shortened in the corpus of the manuscript. Full detail of the methods can be found in tables 2-5, and only key information was retained in the text.

Line 157: it should be referenced to table 2 instead of table 1.

This was corrected.

Line 187 and further: In general, I like the overview on the results in the appendix, it gives a good overview. If possible it would be good to take the tables out of the appendix and place it in the manuscript Then you could also shorten the text of the results part.

We agree with the reviewer’s that the tables provide a nice overview of the results, and, initially, it was our aim to place them in the manuscript. 

However, given the lack of consensus in the literature on the outcome parameters of interest, it is difficult to provide tables that would fit over a page. The tables have indeed a large number of columns and few rows, which compromise their insertion in the text. Therefore, we chose to summarize the results in the paper and to refer to appendix table S2.

Line 187: Please refer to appendix Table S2, also further on in the results section.

We added a reference to the S2 Table in the slope sections, as well as in other sections (l.180-181 of the revised manuscript).

Line 188: replace "speed decreases with the grade of the slope" with "speed decreases with increasing slope"

The sentence was rephrased following the reviewer’s suggestion (l.184).

Line 193: replace "increase with the grade" with "increase with increasing slope"

The sentence was rephrased following the reviewer’s suggestion (l. 188).

Line 215: replace "increase in internal rotation with the slope" with "increase in internal rotation during propulsion on a slope"

This was changed to “increase in internal rotation with increasing slope”.

Line 241: should be referred to table 3 instead of 2.

The reference to the table was corrected. Thank you for spotting this error.

Line 250: before you always called it instrumented wheel instead of handrim dynanometer. Please use one of the labels conistently throughout the manuscript

Thank you for noticing this inconstancy. We replaced three usages (all of them) of “handrim dynamometer” by “instrumented wheel”.

Line 270: "on speed, cycle frequency and duration of push and recovery phase" can be deleted. In general, write more concisely and omit the information that is not really necessary or is a repetition. This makes it easier and more convenient for the reader.

Thank you for your comment which allowed us to increase the readability of our manuscript.

Line 274-275: give reference to the study and Appendix

Reference to the study and to the appendix were added (appendix reference l. 262-263 and study reference l. 268).

Line 296: should be referred to table 4 instead of 3.

This was corrected (l.293).

Line 296: "real" MWC sounds odd, please delete.

This was deleted.

Line 313: add reference to appendix.

A reference to the appendix table was given (l. 306-307).

Line 344: should be referred to table 5 instead of 4.

This was corrected.

Line 365: this is another example of sentences that are not needed, a reference to the appendix is more meaningful.

These sentences were deleted (reference to the appendix l. 352-353).

Discussion

Line 408: "investigated in literature" instead of "from literature"

The sentence was rephrased following the reviewer’s suggestion (l. 392).

Line 409-414: This information belongs to the results, there you already listed how many studies investigated which barrier. It does not have to be mentioned again, especially not with numbers of studies. You might want to say in one sentence what has been studied most.

The sentence was shortened following the reviewer’s suggestion (l. 393-394).

Line 431: safety belts were not used in all of the studies conducted on treadmills

Thank you for your comment. The sentence was changed, and now states that safety belts were sometimes used in studies conducted in treadmills (l. 444-447).

Line 443: Number of studies reporting kinematics and kinetics is for the results section, not for the discussion.

The sentence was shortened following the reviewer’s suggestion.

Line 467: include reference to the study.

The reference to the study was added (l. 451).

References

Reference 1: what kind of a reference is this? if it is a website, please indicate the url.

The reference was corrected to display authors, organisation, and website url.

Reference 11: Title is written twice in the reference

This was corrected.

Figure 1: indicate in the figure why the records (n=1098) they are excluded

The 1098 records were excluded after either a screening on title or abstract. These records were excluded based on our inclusion/exclusion criteria. These criteria are detailed in section (2.2 – Article selection) of the manuscript. “The inclusion criteria were original study or systematic review, study written in English and experimental work. Exclusion criteria were articles about electric wheelchairs, powered-assisted wheelchairs, sports wheelchairs, other propulsion systems than manual handrim, hemiplegia-pattern propulsion”.

It was added to the PRISMA flow-chart that these records were excluded after title or abstract screening.

Reasons detailing why studies were excluded in the full-text screening were also added to the flow-chart.

S2 Table: Check the table, units are not always indicated. For example, in Slope: MWC speed, recovery phase duration, contact angle, sometime it's indicated after the numbers instead of in the the title line (i.e. body kinetics), or both in the title line and after numbers (Peak un-normalized EMG). Please make it consistent.

Thank you for this comment. It is true the reporting of units was inconsistent. All units are now consistently reported. Units for all values except those that are normalized are reported in the title line. Units for values that are normalized are always reported in each cell, because studies did not always use the same normalization criterion, even on the same biomechanical parameter, and therefore had different units for the same parameters.

Appendix

- Contact angle: you define it as "angle distance travelled by the non-dominant hand on the handrim during the push phase". Why is it limited to the non-dominant hand?- Rate of rise: your definition is not correct, not detailed. Is it the mean resultant force divided by the contact time of the whole cycle?

Besides that, in literature there are different definitions:

• taking the derivative of FR with respect to time and then determining the maximum value during the first third of the stroke

• (first) peak of the resultant force, divided by the time to reach the peak.

Therefore, write it more precisely in your definition.

- Fraction of effective force: There are also different definitions used, either FEF = (Mwheel·r−1)·Ftot−1 or Ftan2/Fres2

Thank you for your comment, the definition of these biomechanical parameters was detailed. We presented both definitions for the two latter parameters, and indicated which one was more present in the reviewed articles.

Reviewer 2:

A nicely written systematic review, which presents the work in a very detailed and extensive way. In more detail:

Thank you for the appreciation of our work and your thorough review of the manuscript.

Introduction: the introduction is written well, only the ending could use some finetuning. 

The last sentences of the introduction were moved up in the introduction so as to provide a better ending of the introduction, following the reviewer’s comment.

Methods: Methods section is well presented and only lacks a clear inclusion/exclusion overview.

Inclusion and exclusion criteria were separated in the Methods section following the reviewer’s suggestion.

Results: The results were separated for the four areas (slopes, cross-slopes, curbs and ground type), as well as methods and results.

Regarding the methods on slopes, cross-slopes, curbs and ground type: It is unnecessary to present all results on methods this extensively. Most information can also be found in the presented tables, which already cover a lot of information. Removing parts in these sections might help with the readability of the whole paper.

On the Results on slopes, cross-slopes, curbs and ground type part: Adding subheadings for the subcategories: spatio-temporal, kinematics, kinetics and muscle activity are recommended.

Thank you for your comment. Following your recommendations, the presentation of the reviewed methods in the “Results” section was shortened to only contain information we deemed key for the reader.

Also, as suggested, subheadings were added in the “Results” section for the different biomechanical parameters (spatio-temporal, kinematics, kinetics, muscle activity).

Discussion: In the discussion section a lot of important information is discussed. It feels like a lot of separate interesting discussion points are raised but the structure is a bit lacking. The transitions from one paragraph to another, as well as the general structure should use some finetuning.

Thank you for your comment, we have worked on improving the flow of the “Discussion” section. Paragraphs were re-arranged and subheadings were added to place a greater emphasis on the different key points addressed in the section. We think that this new general structure of the discussion improves its readability compare to the previous draft. 

Conclusion: In this section a lot of information is discussed, which can either be removed or combined with some of the discussion points. Try to get the conclusion short and to the point.

Thank you for your comment and suggestion, the conclusion indeed felt too long. Points addressed in the discussion were either shortened or moved up into the “Discussion” section. The conclusion is now only two paragraphs short.

Smaller general points: The consistency in the use of references, sometimes all references are listed extensively, but there are cases in which some the references are missing, examples are Lines 380-382 and Lines 426-442.

Thank you for noticing these missing references. References were added in both paragraphs as suggested (l. 365, 443-45 of the revised manuscript).

Maybe a figure with the 4 environmental barriers might be a good addition to have a clear overview and explanation of the 4 barriers (cross-slope might be unclear to some naïve readers).

Thank you for your suggestion. Thank you for your comment. Pictures of the recreated environmental barriers were moved from the supplementary materials (S1-3 Figs) to the corpus of the manuscript, and fused in one figure

Abstract:

Page 2, Lines 33-35: Make the conclusion at the end a bit stronger, something similar to: “It is recommended to standardize the procedure, studying a wide set of…”

Following the reviewer’s comment, the concluding sentence of the abstract was rephrased to “It is recommended to standardize the procedure when studying various physical environmental situations and to systematically report MWC configuration(s). Furthermore, a wider set of situations should be studied” (l. 33-36).

Introduction:

Page 3, Lines 45-46: The sentence seems confusing, maybe something similar to: “Despite the improvement of overall accessibility of public areas…”

The sentence was rephrased following the reviewer’s suggestion (l. 45-47).

Page 3, Lines 46: Recommended to start a new paragraph when ICF is introduced

A new paragraph dedicated to the ICF was started (l. 48-60).

Page 3, Lines 61-62: Last sentence of this paragraph feels a bit redundant

Indeed, the last sentence was removed following the reviewer’s comment.

Page 4, Line 75: Remove the word “therefore” in this sentence

The word “therefore” was removed from the sentence.

Page 4, Lines 78-80: This paragraph is unnecessary, the end of the previous paragraph is a more proper ending of the introduction

The sentences of the last paragraph were not deleted as some studies focusing on other means of propulsion were retrieved from the literature and are not discussed in the present review. However, following both reviewer’s comments, they were moved in the Methods section. The introduction now ends on: ” To fill this gap, the purpose of this study was to identify and synthesize data and experimental methods from the literature on the biomechanics of MWC propulsion for various and frequent environmental barriers that are daily encountered by MWC users.” (paragraph moved to l. 82-86).

Methods:

Page 4, Line 83: Change the sentence a bit: “The present study conducted a systematic review to identify…”

The sentence was changed following the reviewer’s comment (l. 82).

Page 5, Lines 105-106: Remove “Appended to this paper…” and add “(S1 table)” add the end of “…(PRISMA) Checklist [24]”

This was changed (l. 108).

Page 5, Lines 106-110: Separate into inclusion/exclusion criteria, inclusion: biomechanical data, experimental work, English languages, exclusion: electric and power assisted wheelchairs etc.

The criteria for article selection into the present systematic review were separated into inclusion and exclusion criteria following the reviewer’s comment (see l. 109-112 of the revised manuscript)

Page 5, Line 111: Remove the word “three” from the sentence

Following the reviewer’s suggestion, “Three” was removed from the sentence as it was already cited earlier in the manuscript that three authors participated in the article selection process.

Page 5, Lines 114-116: Very nice and detailed description of the parameters

Thank you for your comment

Page 6, Line 117: Change towards: “A more detailed definition of the biomechanical parameters can be found in S1 Appendix”

The sentence was changed (l. 119).

Page 6, Line 118-119: Sentence feels a bit redundant

The sentence was moved to the section 2.1 of the manuscript. Indeed, it was a bit out of context in section 2.2 “Article selection” but it seemed relevant to the authors to insist of the methodological aspects of the studies as well as on the obtained results (l. 95-96). 

Results:

Page 6, Lines 129-132: Split the sentence into two

The sentence was split into two sentences and a reference to Table 1 was added following both reviewers’ comments (l. 129-133).

Page 6, Lines 134-137: Cut the sentences a bit, similar to: “Experimental design, acquisition methods and measurement tools differed between studies. The MWC was propelled overground… …ergometer. Kinematics were either recorded…”.

The sentence was cut into three (l. 134-138).

Page 6, Lines 137-138: What is the use of “on the contrary”?

“On the contrary” was removed from the sentence.

Page 6, Lines 140-142: Try to rephrase the sentence a bit, it is unclear what is part of the tables and what of the supplementary material. Maybe similar to: “Table 1 represents the synthesis of all studies, the remaining tables (2-5) summarize the study review for the four categories (slope, cross-slope, curb, ground type). … are appended as supporting information (S2 Table)”.

Following the reviewer’s suggestion, the sentence was rephrased to clarify the organization of the review: “An overview of the retrieved studies is provided in Table 1. A dedicated subsection to each investigated environmental barrier (slope, cross-slope, curb, ground type) synthesizes the experimental methods used in these studies (also reported in Tables (2-5) ) as well as the obtained biomechanical results with summarizing the experimental method. A compilation of the studies' numerical results is appended to this document as supplementary material (S2 Table).” (l. 139-143).

Page 18-19, Methods on slopes: See general comment

As answered before in general comments, the presentation of the experimental methods of the reviewed studies was shortened. Full detail of the methods can be found in table 2, and only key information was retained in the text.

Page 19, Line 188-189: Sentence can be removed (“Results on cycle frequency…) and combined with the next sentence.

This sentence was removed.

Page 19, Lines 191-194: Try to shorten the sentence: “…increase with the grade in SCI subjects, but unaffected with AB subjects.”

The sentence was shortened as suggested.

Page 19: Lines 201-203: What kind of differences were found?

Thank you for this comment, we indeed did not detail which differences were found. A sentence detailing these differences was added to the manuscript: ”Contact angle was higher on the same slope when experimenting on AB subjects, and seemed to remain constant with different grades of slope in AB subjects, whilst contact angle tended to decrease with increasing slope in MWU.” (l. 198-200).

Page 20: Line 212: Remove the part “in the study where they were reported”.

This part was removed.

Page 20: Lines 219-220: Remove the part: “…in all studies reporting these data”.

This part was removed.

Page 21: Line 233: Change to past sense

The verb “give” was changed to the past tense, thank you for spotting that error.

Page 21-22, Methods on cross-slopes: See general comments

See general comments for response: the review of the studies’ methods was shortened in the text. While tables still give full detail on the methods of the reviewed studies, the text now holds only key information.

Page 22: Lines 266-267: Is an interpretation sentence, better be part of the discussion

Thank you for your suggestion. The sentence was moved into the discussion section (4.1) (l. 395-397 of the revised manuscript).

Page 22, Lines 270-274: Try to shorten this sentence, parts at the beginning of the sentence can be removed.

The sentence was rephrased (l. 265-268).

Page 23-24, Methods on curbs: See general comments

See general comments for response: the review of the studies’ methods was shortened in the text. While tables still give full detail on the methods of the reviewed studies, the text now holds only vital information.

Page 23, Line 296: Remove the word “real”

This word was removed.

Page 25-26, Methods on ground types: See general comments

See general comments for response: the review of the studies’ methods was shortened in the text. While tables still give full detail on the methods of the reviewed studies, the text now holds only vital information.

Page 27, Lines 380-382: The references are missing

References were added (l. 365).

Discussion:

Page 27-28, Lines 401-406: This paragraph only repeats the aims, without actually giving any relevant information, combining some of this information with the next paragraph might be a solution.

Thank you for your comment. The paragraph was shortened to avoid repetition and allows for a better introduction of the review’s focus on experimental design issues (l. 386-388).

Page 28, Lines 417-425: This part can be shortened a bit, there is some repetition, especially with the last sentence.

This part of the paragraph was split into two, creating one more paragraph to present our point better. The penultimate sentence of the part was changed to get rid of repetition (l. 406-415).

Page 29, Lines 428-430: Future and past sense are both used in this sentence

The sentence was changed so the verbs are conjugated accordingly (l. 417-418).

Page 29-30, Lines 443-454: The paragraph starts with summarizing the number of articles investigating kinematics/kinetics. Despite main focus of this paragraph seems on the differences in acquisition methods. Try to get the most relevant information at the start/end of the paragraph.

Thank you for this comment. Along with rewiever 1’s suggestions, the first sentence of the paragraph was shortened to skip the summary of articles investigating kinematics/kinetics (l. 471-472).

Page 30, Lines 469-476: This feels like a separate paragraph, since the main focus of this paragraph was on the use of instrumented measurement wheels on both side of the MWC.

Indeed, not starting a new paragraph felt odd. This was corrected and a new paragraph was created discussing joint coordinate systems usage (new paragraph l. 497-507).

Page 31, Lines 486-490: It is unclear why this sentence is presented in between brackets, would make the argument stronger to present it as an example.

Thank you for your remark, the part in the brackets was moved out of brackets to be used as an example (l. 529-531).

Page 32, Lines 505-508: Try to combine these sentences into one strong sentence.

These two sentences were combined into one (l.466-468 of the revised manuscript).

Page 32, Lines 509-516: This paragraph does not add too much to the whole review and could, feels like lose paragraph at the end of the discussion, see also general comments

The paragraph was moved up in the discussion, where it was more relevant (section 4.3) and was slightly reformulated (l. 508-517). 

Conclusion:

Page 32-34: A conclusion is preferably short and to the point. The conclusion now consists of around five paragraphs in which also a lot of recommendations are given. Some of these recommendations are already discussed previously and don’t necessarily strengthen the paper.

Thank you for your comment. Following your recommendations, the conclusion was shortened and is now straighter to the point.

Page 32, Lines 519-524: The first paragraph summarizes/repeats what has already been repeated multiple times

The first paragraph was deleted.

Page 33, Line 526: This is one of the main lines to be used for the conclusion.

Page 33, Lines 530-536: This all can be said in one sentence in the discussion part.

The paragraph was moved to the discussion section and split into different sentences adding weight to their respective new paragraphs.

Page 33, Lines 537-540: Interesting point, also combine with some of the paragraphs in the discussion part (Future recommendation)

The paragraph was moved in the discussion, in the section 4.1 discussing the environmental barriers studied in the retrieved literature (l. 406-409).

Page 33, Lines 541-543: Repeating the fact that cross-slopes and curbs should be studies, but now focused on asymmetrical propulsion.

This paragraph indeed felt like a repetition of what was already discussed previously, for this reason, it was deleted.

Page 33-34, Lines 544-551: This seems like one of the main parts of the conclusion of the paper

This paragraph is now the conclusion’s and our manuscript’s end (l. 545-552 of the revised manuscript).

---

## [Decision Letter · Decision Letter 1]

7 Apr 2022

PONE-D-21-13523R1Manual wheelchair biomechanics while overcoming various environmental barriers: a systematic reviewPLOS ONE

Dear Dr. Rouvier,

Thank you for submitting your manuscript to PLOS ONE. After careful consideration, we feel that it has merit but does not fully meet PLOS ONE’s publication criteria as it currently stands. Therefore, we invite you to submit a revised version of the manuscript that addresses the points raised during the review process.

We look forward to receiving your revised manuscript.

Kind regards,

Lucas van der Woude

Academic Editor

PLOS ONE

Journal Requirements:

Reviewers' comments:

Reviewer's Responses to Questions

**Comments to the Author**

1. If the authors have adequately addressed your comments raised in a previous round of review and you feel that this manuscript is now acceptable for publication, you may indicate that here to bypass the “Comments to the Author” section, enter your conflict of interest statement in the “Confidential to Editor” section, and submit your "Accept" recommendation.

Reviewer #1: (No Response)

Reviewer #2: (No Response)

2. Is the manuscript technically sound, and do the data support the conclusions?

Reviewer #1: Yes

Reviewer #2: Yes

3. Has the statistical analysis been performed appropriately and rigorously? 

Reviewer #1: N/A

Reviewer #2: Yes

4. Have the authors made all data underlying the findings in their manuscript fully available?

Reviewer #1: Yes

Reviewer #2: Yes

5. Is the manuscript presented in an intelligible fashion and written in standard English?

Reviewer #1: Yes

Reviewer #2: Yes

6. Review Comments to the Author

Reviewer #1: The authors have addressed my comments well and the manuscript is now much clearer and easier to read.

I only have some small comments to this revised version:

Abstract:

In my view the conclusion stated in the abstract does not reflect the conclusion of your whole paper. Please include all recommendatios given in the conclusion section (i-v), or state it more general. Furthermore, do you suggest to standardize the measurements or the reporting of it?

Methods:

Line 107: You refer to the PRISMA checklist when describing article selection. The prisma Checklist does not specify how to select articles but specifies what should be reported in a systematic review. So better refer to figure 2 or rewrite the first sentence.

Line 110: You mention systematic review as a inclusion criteria, but another inclusion criteria is experimental work. If experimental work is a inclusion criteria, systematic reviews (as mentioned in the line above) will not be included.

Results:

Figure 2: In the description of figure 2 you indicate classification, but this is not addressed in figure 2.

Line 286: Please indicate whether EMG was measured on both sides or just on up-/ or downhill side.

Line 344: include "wheel" after instrumented

Discussion:

Line 440: include "due" in front of to.

Conclusion:

Line 548-553: here you state what you recommend for further studies. Is this what you miss in the current studies, or is this all you would like to have reported in studies? And what for example about the participant characteristics?

Reviewer #2: Comments to editor:

The authors addressed all essential comments previously made. Only some minor revisions are necessary, after which the paper is recommended to be accepted.

General comments:

Thank you for addressing all comments and the changes made to the manuscript. The changes have improved the readability of the manuscript. Only some minor adjustments are recommended.

A small note: Sometimes a tab is used to begin a new paragraph, sometimes not, there is no real consistency throughout the whole paper.

Specific comments:

Abstract:

Page 2, Lines 33-36: It would be recommended to make 1 strong final sentence: “It is recommended to standardize the procedure when studying various physical environmental situations, systematically report MWC configuration(s) and study a wider set of situations.”

Introduction:

The introduction now indeed ends with a logical flow towards the purpose of the study, well done.

Methods:

Page 5, Lines 109-112: The exclusion criteria are now nicely stated, only the inclusion criteria are still a bit too general (English language, original study, experimental work). It would be recommended to also add something similar to: “Studies investigating slope, cross-slope, curb and ground type in manual wheelchair users.”

Results:

Removing parts on the ‘methods’ section for all barriers improved the readability of the results part of the manuscript. Adding subheading really helped for the structure as well, great work.

Page 19, Line 180-182: These two sentences can be removed or moved a bit up, it is already previously stated on Page 7, Line 144 that the numerical results can be found in S2 Table. Now the same sentence is stated for all barriers on Page 22, Lines 263-264, Page 24, Lines 307-308 and Page 26, Lines 353-354. Stating that information once around line 144 will probably be enough, similar to: ‘Supplementary material on the detailed results on all environmental barriers (slopes, curbs etc.) can be found in S2 Table.’.

Page 22, Line 265: The subheading ‘3.2.2.1. Spatio-temporal parameters’ is too big, should be similar to the ‘3.2.2.2. Joint Kinematics’.

Discussion

Page 27, Lines 387-389: This introductory statement might not necessarily be needed, in the next paragraph it is also summarized that environmental barriers were investigated.

Page 28, Lines 411-422: One nice combined paragraph can be made out of these three separate paragraphs with the last one only being one sentence.

Page 31, Lines 483-497: This has become a rather long paragraph to only state one thing: ‘Instrumented wheels should be used on both sides’. Especially lines 490-495 are repetitions of the first sentences of the paragraph.

Page 32, Lines 509-511: Which studies are you referring to?

Page 32, Line 511: ‘… might have resulted in significant uncertainties’.

Conclusion:

The conclusion is a lot more shortened and to the point, well done.

Page 33, Line 538: The first sentence is a bit unclear, seems like there is missing some information.

Page 33, Line 545: ‘… parameters retrieved from the relevant literature.’

7. PLOS authors have the option to publish the peer review history of their article (what does this mean?). If published, this will include your full peer review and any attached files.

Reviewer #1: **Yes: **Ursina Arnet

Reviewer #2: **Yes: **Riemer Vegter & Thomas Rietveld

---

## [Author Response · Author response to Decision Letter 1]

25 Apr 2022

Dear reviewers,

Thank you for your encouraging comments while reviewing the revised manuscript and for allowing us to keep improving its content. 

Please find below how we have addressed each of the reviewers’ comment. The reviewers’ comments are reported in black, whereas the answers are reported in blue.

A version of the revised manuscript has also been uploaded. Changes are highlighted in yellow in the revised version of the manuscript.

Kindest regards,

Théo Rouvier, on behalf of all the authors of the manuscript entitled "Manual wheelchair biomechanics while overcoming various environmental barriers: a systematic review”

Reviewer 1:

The authors have addressed my comments well and the manuscript is now much clearer and easier to read.

I only have some small comments to this revised version:

Thank you for your endearing comments. We will try to take the latter comments into account as best as we can.

Abstract:

In my view the conclusion stated in the abstract does not reflect the conclusion of your whole paper. Please include all recommendations given in the conclusion section (i-v), or state it more general. Furthermore, do you suggest to standardize the measurements or the reporting of it?

Thank you for your comment. We added the five recommendations made in the conclusion to the abstract. These recommendations include suggestions to both standardize the measurements (ii) and the reporting of it (i, iii, iv, v).

Methods:

Line 107: You refer to the PRISMA checklist when describing article selection. The prisma Checklist does not specify how to select articles but specifies what should be reported in a systematic review. So better refer to figure 2 or rewrite the first sentence.

Indeed, we should not have referred to the prisma checklist. The sentence now refers to figure 2, which was moved up in the manuscript to appear after this paragraph.

Line 110: You mention systematic review as a inclusion criteria, but another inclusion criteria is experimental work. If experimental work is a inclusion criteria, systematic reviews (as mentioned in the line above) will not be included.

Thank you for your comment. This error was fixed by modifying the phrasing of the inclusion criteria, which now state “features experimental results” instead of “experimental work”

Results:

Figure 2: In the description of figure 2 you indicate classification, but this is not addressed in figure 2.

Thank you for your comment. This was from an earlier draft of the figure. Classification was removed from the figure’s title.

Line 286: Please indicate whether EMG was measured on both sides or just on up-/ or downhill side.

EMG was measured only on the left side. This precision was added in the presentation of the methods and results of muscle activity in cross-slopes.

Line 344: include "wheel" after instrumented

This was corrected.

Discussion:

Line 440: include "due" in front of to.

This was corrected.

Conclusion:

Line 548-553: here you state what you recommend for further studies. Is this what you miss in the current studies, or is this all you would like to have reported in studies? And what for example about the participant characteristics?

Thank you for this question. These recommendations are based on what we miss in most studies of the literature, but also what we would like see reported in future studies. For example, some studies -but not all- reported barrier’s lengths and other characteristics, and we deem it is important that this practice be generalized. On the other hand, no reviewed study reported measurement uncertainties or model reconstruction errors, and we think it would be a truly interesting metric for researchers trying to compare their results with the state of the art. Participant characteristics are also a very interesting metric, but we feel that the state of the art has reached closer to a consensus, hence why we did not include them in our recommendations.

Reviewer 2: 

Thank you for addressing all comments and the changes made to the manuscript. The changes have improved the readability of the manuscript. Only some minor adjustments are recommended.

A small note: Sometimes a tab is used to begin a new paragraph, sometimes not, there is no real consistency throughout the whole paper.

Thank you for your encouraging comments. Tabulations are now systematically used at the start of the first paragraph of a section.

Abstract:

Page 2, Lines 33-36: It would be recommended to make 1 strong final sentence: “It is recommended to standardize the procedure when studying various physical environmental situations, systematically report MWC configuration(s) and study a wider set of situations.”

Thank you for your comment. Alongside reviewer 1’s comments, we decided to end the abstract by the five recommendations listed in the conclusion. The sentence “Furthermore a wider set of situations should be studied” that was used to close the abstract was moved up to let our recommendations close the abstract.

Introduction:

The introduction now indeed ends with a logical flow towards the purpose of the study, well done.

Thank you very much for your comment.

Methods:

Page 5, Lines 109-112: The exclusion criteria are now nicely stated, only the inclusion criteria are still a bit too general (English language, original study, experimental work). It would be recommended to also add something similar to: “Studies investigating slope, cross-slope, curb and ground type in manual wheelchair users.”

Thank you for your comment. We changed Inclusion criteria and replaced “experimental work” by “features experimental results on slopes, cross-slopes, curbs, and ground types during MWC locomotion”.

Results:

Removing parts on the ‘methods’ section for all barriers improved the readability of the results part of the manuscript. Adding subheading really helped for the structure as well, great work.

Thank you for your comment.

Page 19, Line 180-182: These two sentences can be removed or moved a bit up, it is already previously stated on Page 7, Line 144 that the numerical results can be found in S2 Table. Now the same sentence is stated for all barriers on Page 22, Lines 263-264, Page 24, Lines 307-308 and Page 26, Lines 353-354. Stating that information once around line 144 will probably be enough, similar to: ‘Supplementary material on the detailed results on all environmental barriers (slopes, curbs etc.) can be found in S2 Table.’.

Thank you for your comment. Duplicates were removed, and the reference to Table S2 is now only listed in the overview of general results.

Page 22, Line 265: The subheading ‘3.2.2.1. Spatio-temporal parameters’ is too big, should be similar to the ‘3.2.2.2. Joint Kinematics’.

This was corrected.

Discussion

Page 27, Lines 387-389: This introductory statement might not necessarily be needed, in the next paragraph it is also summarized that environmental barriers were investigated.

The introductory statement was deleted.

Page 28, Lines 411-422: One nice combined paragraph can be made out of these three separate paragraphs with the last one only being one sentence.

Thank you for your comment. The three paragraphs were merged into one

Page 31, Lines 483-497: This has become a rather long paragraph to only state one thing: ‘Instrumented wheels should be used on both sides’. Especially lines 490-495 are repetitions of the first sentences of the paragraph.

Thank you for your comment. The paragraph indeed felt long. We shortened the paragraph to restrain from repeating ourselves, and focused the latter half of the paragraph on the fact that the only study that investigated cross-slopes using two instrumented wheels, reported results on the dominant and non-dominant sides of the user without expressing which one was downhill and uphill, which is also an important factor in the biomechanics of cross-slope locomotion

Page 32, Lines 509-511: Which studies are you referring to?

We added the precision as to which studies we are referring to: all studies presented in this review.

Page 32, Line 511: ‘… might have resulted in significant uncertainties’.

This was corrected

Conclusion:

The conclusion is a lot more shortened and to the point, well done.

Thank you for your comment.

Page 33, Line 538: The first sentence is a bit unclear, seems like there is missing some information.

The first sentence was rephrased: “This review highlighted discrepancies in focus given to each environmental situation in the literature”.

Page 33, Line 545: ‘… parameters retrieved from the relevant literature.’

This was corrected

---

## [Editor Report · Decision Letter 2]

26 May 2022

Manual wheelchair biomechanics while overcoming various environmental barriers: a systematic review

PONE-D-21-13523R2

Dear Dr. Rouvier,

We’re pleased to inform you that your manuscript has been judged scientifically suitable for publication and will be formally accepted for publication once it meets all outstanding technical requirements. 

Kind regards,

Lucas van der Woude

Academic Editor

PLOS ONE

---

## [Editor Report · Acceptance letter]

31 May 2022

PONE-D-21-13523R2 

Manual wheelchair biomechanics while overcoming various environmental barriers: a systematic review 

Dear Dr. Rouvier:

I'm pleased to inform you that your manuscript has been deemed suitable for publication in PLOS ONE. Congratulations! Your manuscript is now with our production department. 

Kind regards, 

on behalf of

Professor Lucas van der Woude 

Academic Editor

PLOS ONE